# On the use of GRACE normal equation of intersatellite tracking data for estimation of soil moisture and groundwater in Australia

**Natthachet Tangdamrongsub** [1], **Shin-Chan Han** [1], **Mark Decker** [2], **In-Young Yeo** [1], **Hyungjun Kim** [3]

[1] School of Engineering, University of Newcastle, Callaghan, New South Wales, Australia
[2] ARC Centre of Excellence for Climate System Science, University of New South Wales, Sydney, New South Wales, Australia
[3] Institute of Industrial Science, the University of Tokyo, Tokyo, Japan

## Abstract

An accurate estimation of soil moisture and groundwater is essential for monitoring the availability of water supply in domestic and agricultural sectors. In order to improve the water storage estimates, previous studies assimilated terrestrial water storage variation ($\Delta TWS$) derived from Gravity Recovery and Climate Experiment (GRACE) into land surface models. However, the GRACE-derived $\Delta TWS$ was generally computed from the high level products (e.g., time-variable gravity fields, i.e., Level 2, and land grid from the Level 3 product). The gridded data products are subjected to several drawbacks such as signal attenuation and/or distortion caused by posteriori filters, and a lack of error covariance information. The post-processing of GRACE data might lead to the undesired alteration of the signal and its statistical property. This study uses the GRACE least-squares normal equation data to exploit the GRACE information rigorously and negate these limitations. Our approach combines the GRACE's least-squares normal equation (obtained from ITSG-Grace2016 product) with the results from the Community Atmosphere Land Exchange (CABLE) model to improve soil moisture and groundwater estimates. This study demonstrates, for the first time, an importance of using the GRACE raw data. The GRACE-combine (GC) approach is developed for optimal least-squares combination and the approach is applied to estimate the soil moisture and groundwater over 10 Australian river basins. The results are validated against the satellite soil moisture observation and the in-situ groundwater data. Comparing to CABLE, we demonstrate the GC approach delivers evident improvement of water storage estimates, consistently from all basins, yielding better agreement at seasonal and inter-annual time scales. Significant improvement is found in groundwater storage while marginal improvement is observed in surface soil moisture estimates.

## 1. Introduction

The changes of Terrestrial Water Storage ($\Delta TWS$) derived from the Gravity Recovery And Climate Experiment (GRACE) data products have been used in the last decade to study global water resources, including groundwater depletion in India and Middle East (Rodell et al., 2009; Voss et al., 2013), water storage accumulation in Canada (Lambert et al., 2013), flood-influenced water storage fluctuation in Cambodia (Tangdamrongsub et al., 2016). The gravity data obtained from GRACE satellites are commonly processed and released in three different product levels (L) that increase in the amount of processing, L1B – satellite tracking data (e.g., Wu et al., 2006), L2 – global gravitational Stokes coefficients (e.g., Bettadpur,

2012), and L3 – global grids (e.g., Landerer and Swenson, 2012). The original (L1B)
GRACE information is inevitably altered or sheered due to data processing and successive
post-processing filterings, because the error covariance information is not propagated through
each post-processing step.
The GRACE-derived $\Delta TWS$ has been computed widely from the higher-level products (e.g.,
L2 and L3) on which various ad hoc post-processing filters were applied (e.g., Gaussian
smoothing filter (e.g., Jekeli, 1981), destripe filter (e.g., Swenson and Wahr, 2006)). $\Delta TWS$
obtained from these filters lacks proper error covariance information and is attenuated and
distorted. To overcome the signal attenuation in GRACE high-level products, empirical
approaches have been developed, including the application of scale factors computed from
land surface models (Landerer and Swenson, 2012) to the GRACE L3 products. GRACE
uncertainty in high level product is usually unknown or assumed. For example, Zaitchik et al.
(2008) derived empirically a global average uncertainty that is variable depending on choices
of post-processing filters (Sakumura et al., 2014). Furthermore, GRACE error and sensitivity
is dependent on latitudes due to the orbit convergence toward poles (Wahr et al., 2006) and
any post-processing filters will alter the GRACE data and their error information. Rigorous
statistical error information is of equal importance to derivation of $\Delta TWS$ for data
assimilation and model calibration (Tangdamrongsub et al., 2017; Schumacher et al., 2016,
2018). $\Delta TWS$ and its uncertainty estimates should be formulated directly from L1B data
considering the complete statistical information.
The GRACE information is not fully exploited in many studies. For example, groundwater
storage variation ($\Delta GWS$) is often computed by subtracting the soil moisture variation ($\Delta SM$)
component simulated by the land surface model from GRACE-derived $\Delta TWS$ data (Rodell et
al., 2009, Famiglietti et al., 2011), assuming the model $\Delta SM$ is error-free. This may result in
the inaccurate $\Delta GWS$ and the associated error estimate as the uncertainties of observation and
of the land surface model outputs are neglected in the combination (or regression) of two
noisy data (e.g., Long et al., 2016). In data assimilation, the GRACE uncertainty is often
derived empirically, not necessarily reflecting the actual GRACE error characteristics (e.g.,
Zaitchik et al., 2008; Tangdamrongsub et al., 2015; Tian et al., 2017). For example, Girotto et
al. (2016) used L3 product and showed that it was necessary to adjust GRACE observation
and its uncertainty in order to make their water storage estimates more accurate. Similarly,
Tian et al. (2017) reported the need of applying a scale factor to GRACE uncertainty (from
mascon product) in their GRACE assimilation process. It is apparent that the use of post-
processed GRACE products often requires data tuning, leading possibly to an integration of
the altered gravity information into the data assimilation system. Some recent studies began
to employ the full variance-covariance information in the data assimilation scheme to
enhance the quality of the estimates (Eicker et al., 2014, Schumacher et al., 2016;
Tangdamrongsub et al., 2017; Khaki et al., 2017 a,b).
This study aims to use the GRACE information of $\Delta TWS$ measurement directly from the
least-squares normal equation data. The approach optimally combines the GRACE's normal
equations with the model simulation results from the Community Atmosphere Land
Exchange (CABLE, Decker, 2015) to improve $\Delta SM$ and $\Delta GWS$ estimates. The proposed
approach presents three main advantages. Firstly, one can exploit the full GRACE signal and
error information by using the normal equation data sets. Secondly, the approach is
developed for optimal least-squares combination (e.g., Ramillien et al., 2004), which
maximizes the model and observation strength while simultaneously supressing their
weaknesses. Finally, the method bypasses empirical, multiple-step post-processing filters.
The main objective of this study is to present the GRACE-combined (GC) approach to
improve the model estimated $\Delta SM$ and $\Delta GWS$ at regional scales. We demonstrate our
approach applied to 10 Australian river basins (Fig. 1a). One advantage of the study area is
that the state vector can be simply defined by $\Delta SM$ and $\Delta GWS$ as other hydrological
components (e.g., snow, glacier) are negligible. We validate the top layer of $\Delta SM$ estimates
against the satellite soil moisture observation (the Advanced Microwave Scanning
Radiometer aboard EOS (AMSR-E), Njoku et al., 2003) over all 10 basins and the $\Delta GWS$
estimates against the in-situ groundwater data available over Queensland and Victoria (Fig.
1b, 1c).
This paper is outlined as follows: Firstly, the derivation of GC approach is presented in Sect.
2 while the description of GRACE data processing, including the use of GRACE normal
equation, is given in Sect. 3. Secondly, the CABLE modelling is outlined in Sect. 4. This
includes the derivation of model uncertainty based on the quality of precipitation data and the
model parameter inputs. The processing of validation data is also described in Sect. 4.
Thirdly, Sect. 5 presents the result of $\Delta SM$ and $\Delta GWS$ estimates and comparison to in-situ
data. The long-term trends in the Australian mass variation over the last 13 years is also
investigated in this section.

**2. A method of combining GRACE L1B data with land surface model outputs**
The statistical information of $\Delta TWS$ computed from a land surface model can be written as:
$$\widetilde{\boldsymbol{h}} = \boldsymbol{h} + \boldsymbol{\epsilon}; \boldsymbol{\epsilon} \sim \mathcal{N}(\boldsymbol{0}, \mathbf{C}), \qquad (1)$$
where $\boldsymbol{h}$ is the "truth" (unknown) model state vector while $\widetilde{\boldsymbol{h}}$ is the calculated state vector
characterized with the model error $\boldsymbol{\epsilon}$. The model error is assumed to have zero mean and
covariance $\mathbf{C}$.
The term $\boldsymbol{h}$ is used to represent a vector including global $\Delta TWS$ grid, and terms with a
subscript $R$ (e.g., $\boldsymbol{h_R}$, $\mathbf{C_R}$) is used to represent only a regional set of $\Delta TWS$ (for example, in
Australia). As such, the observation equation over a region can be rewritten as:
$$\widetilde{\boldsymbol{h}}_R = \boldsymbol{h}_R + \boldsymbol{\epsilon}; \boldsymbol{\epsilon} \sim \mathcal{N}(\boldsymbol{0}, \mathbf{C_R}). \qquad (2)$$
As soil moisture and groundwater are the major components of $\Delta TWS$ in Australia (surface
water storage being insignificant), the vector $\boldsymbol{h_R}$ can be defined as:
$$\boldsymbol{h_R} = [\Delta \boldsymbol{SM}_{top} \quad \Delta \boldsymbol{SM}_{rz} \quad \Delta \boldsymbol{GWS}]^T, \qquad (3)$$
where $\Delta \boldsymbol{SM}_{top}, \Delta \boldsymbol{SM}_{rz}, \Delta \boldsymbol{GWS}$ represent the vectors of top (surface) soil moisture, root zone
soil moisture, and groundwater storage variations, respectively.
A least-squares normal equation of GRACE can be written as:
$$\mathbf{N}\,\boldsymbol{x} = \boldsymbol{c} \qquad (4)$$
Where **N** is a normal matrix, $\boldsymbol{x}$ contains the spherical harmonic coefficients (SHC) of the
geopotential, and $\boldsymbol{d}$ is the normal vector. In this study, **N** and $\boldsymbol{c}$ can be obtained from the
ITSG-Grace2016 products (Mayer-Gürr et al, 2016;
https://www.tugraz.at/institute/ifg/downloads/gravity-field-models/itsg-grace2016, see more
details in Sect. 3.1). Eq. (4) can be written in terms of $\boldsymbol{h}$ as follows (see Appendix A for the
derivation):

$$(\mathbf{H}^T\mathbf{Y}^T\mathbf{N}\mathbf{Y}\mathbf{H})\widehat{\boldsymbol{h}} = \mathbf{H}^T\mathbf{Y}^T\boldsymbol{c} \tag{5}$$

where **Y** converts $\Delta TWS$ to geopotential coefficients considering the load Love numbers
(e.g., Wahr et al., 1998) and **H** is the operational matrix converting $\Delta \boldsymbol{SM}_{top}$, $\Delta \boldsymbol{SM}_{rz}$, and
$\Delta \boldsymbol{GWS}$ to $\Delta TWS$. Eq. (5) is based on the assumption that the GRACE orbital perturbation is a
result of $\Delta TWS$ variation on the surface. If $M$ is the number of model grid cells, $N_{max}$ is the
maximum degree of the geopotential coefficients, and $L=(N_{max}+1)^2–4$ is the number of
geopotential coefficients from GRACE, the dimension of **Y**, **H**, and $\boldsymbol{h}$ are $L \times M$, $M \times 3M$, and
$3M \times 1$, respectively. Note that, Eq. (5) is defined with the global grid of $\boldsymbol{h}$. For a regional
application, Eq. (5) can be modified as:

$$\begin{bmatrix}\mathbf{H}_R^T\mathbf{Y}_R^T \mid \mathbf{H}_o^T\mathbf{Y}_o^T\end{bmatrix} \mathbf{N} \begin{bmatrix}\mathbf{Y}_R\mathbf{H}_R \\ \mathbf{Y}_o\mathbf{H}_o\end{bmatrix} \begin{bmatrix}\widehat{\boldsymbol{h}}_R \\ \widehat{\boldsymbol{h}}_o\end{bmatrix} = \begin{bmatrix}\mathbf{H}_R^T\mathbf{Y}_R^T \mid \mathbf{H}_o^T\mathbf{Y}_o^T\end{bmatrix} \boldsymbol{c}, \tag{6}$$

where the subscript $R$ indicates the grid $\Delta TWS$ only in a region of interest, and $o$ for the rest
of the globe. If the number of the model grid cells associated with $R$ is $J$ and that of the
outside cells is $M–J$. As such, the dimensions of $\mathbf{Y}_R$, $\mathbf{H}_R$, $\widehat{\boldsymbol{h}}_R$, $\mathbf{Y}_o$, $\mathbf{H}_o$, $\widehat{\boldsymbol{h}}_o$ are $L \times J$, $J \times 3J$, $3J \times 1$,
$L \times (M–J)$, $(M–J) \times 3(M–J)$, $3(M–J) \times 1$, respectively. The dimension of **N** and $\boldsymbol{c}$ remain
unchanged, since they are essentially from the normal equations of the original GRACE L1B
data (to be discussed in the following section).
From Eq. (6), the normal equations associated with $\Delta TWS$ in the region of interest can then
be written as

$$\mathbf{H}_R^T\mathbf{Y}_R^T\mathbf{N}\mathbf{Y}_R\mathbf{H}_R\widehat{\boldsymbol{h}}_R = \mathbf{H}_R^T\mathbf{Y}_R^T\boldsymbol{c} - \mathbf{H}_R^T\mathbf{Y}_R^T\mathbf{N}\mathbf{Y}_o\mathbf{H}_o\widehat{\boldsymbol{h}}_o \tag{7}$$

or

$$\mathbf{N}_R\widehat{\boldsymbol{h}}_R = \boldsymbol{c}_R \tag{8}$$

where $\mathbf{N}_R = \mathbf{H}_R^T\mathbf{Y}_R^T\mathbf{N}\mathbf{Y}_R\mathbf{H}_R$ and $\boldsymbol{c}_R = \mathbf{H}_R^T\mathbf{Y}_R^T\boldsymbol{c} - \mathbf{H}_R^T\mathbf{Y}_R^T\mathbf{N}\mathbf{Y}_o\mathbf{H}_o\widehat{\boldsymbol{h}}_o$. As seen, Eq. (7) is the
regional representation of Eq. (5) where only the grid cells inside the study region are used,
while the contribution from the grid cells outside the region needs to be removed or
corrected. Combining the normal equation of Eq. (2) and Eq. (8), the optimal combined
solution of $\widehat{\boldsymbol{h}}_R$ can be resolved as follows:

$$\widehat{\boldsymbol{h}}_R = \left(\mathbf{C}_R^{-1} + \mathbf{N}_R\right)^{-1}\left(\mathbf{C}_R^{-1}\widetilde{\boldsymbol{h}}_R + \boldsymbol{c}_R\right) \tag{9}$$

The computation of model covariance matrix $\mathbf{C}_R$ will be discussed in Sect. 4.2. The posteriori
covariance of $\widehat{\boldsymbol{h}}_R$ can be estimated as follows:

$$\widehat{\boldsymbol{\Sigma}} = (\mathbf{C}_R^{-1} + \mathbf{N}_R)^{-1}, \tag{10}$$

and the uncertainty estimate of $\widehat{\boldsymbol{h}}_R$ is simply calculated as:

$$\boldsymbol{\sigma}_{\widehat{h}} = \sqrt{diag(\widehat{\boldsymbol{\Sigma}})}, \tag{11}$$

where $diag()$ represents the diagonal element of the given matrix.

## 3. GRACE data

### 3.1 GRACE least-squares normal equations

In this study, the least-squares normal equations are obtained from the ITSG-Grace2016
products between January 2003 and March 2016. All L1B data including KBR inter-satellite
tracking data, attitude, accelerometer, GPS based kinematic orbit data and AOD1B
corrections are reduced in terms of the normal equations. These data products are usually
used to compute the Earth's geopotential field to the maximum harmonic degree and order of
90, or at a spatial resolution of ~220 km. The products contain the information of the normal
matrix $\mathbf{N}$ and the vector $\boldsymbol{c}$ (as shown in Eq. (4)) as well as the a-priori time-varying gravity
field coefficients predicted with the GOCO05s solution (Mayer-Gürr et al., 2015). Note that
the solution of the ITSG-Grace2016 normal equation is the anomalous geopotential
coefficient vector ($\Delta\boldsymbol{x}$), which is referenced to the a-priori time-varying gravity field ($\boldsymbol{x_0}$),
through:

$$\mathbf{N}\,\Delta\boldsymbol{x} = \boldsymbol{d} \tag{12}$$

where $\boldsymbol{d}$ and $\boldsymbol{x_0}$ are given. To obtain a complete gravity field variation between the study
period ($\boldsymbol{x}$ term in in Eq. (4)), the a-priori time-varying gravity field, $\boldsymbol{x_0}$ is firstly restored to
Eq. (12), and the mean gravity field ($\overline{\boldsymbol{x}}_0$) computed from all $\boldsymbol{x_0}$ between January 2003 and
March 2016 is then removed as follows:

$$\mathbf{N}\,(\Delta\boldsymbol{x} + \boldsymbol{x_0} - \overline{\boldsymbol{x}}_0) = \boldsymbol{d} + \boldsymbol{N}(\boldsymbol{x_0} - \overline{\boldsymbol{x}}_0) \tag{13}$$

$$\mathbf{N}\,\boldsymbol{x} = \boldsymbol{d} + \boldsymbol{N}(\boldsymbol{x_0} - \overline{\boldsymbol{x}}_0) \tag{14}$$

Therefore, in Sect. 2 (e.g., Eq. (5)), the matrix $\mathbf{N}$ remains unchanged while the vector $\boldsymbol{c}$ can
be simply replaced by $\boldsymbol{c} = \boldsymbol{d} + \boldsymbol{N}(\boldsymbol{x_0} - \overline{\boldsymbol{x}}_0)$.

### 3.2 GRACE-derived $\Delta TWS$ products

Three monthly GRACE-derived $\Delta TWS$ products are also used, the ITSG-Grace2016 DDK5
solution (ITSG-DDK5 for short, http://icgem.gfz-potsdam.de/series/99_non-iso/ITSG-
Grace2016), the CNES/GRGS Release 3 (RL3) (GRGS for short, Lemoine et al., 2015;
http://grgs.obs-mip.fr/grace/variable-models-grace-lageos/grace-solutions-release-03) and the
JPL RL05M mascon-CRI version 2 product (mascon for short, Watkins et al., 2015; Wiese et
al., 2016; http://grace.jpl.nasa.gov/data/get-data/jpl_global_mascons). The ITSG-DDK5
product is the post-processed version of the ITSG L2 solution where the non-isotropic filter
DDK5 (Kusche et al., 2009) is applied. The DDK5 solution is empirically selected here to be
a good balance between the over-smoothed (e.g., DDK1) and noisy (e.g., DDK8) solutions.
The GRGS solution provides $\Delta TWS$ at $1^\circ \times 1^\circ$ globally, derived from the Earth's geopotential
coefficients up to the maximum degree and order 80, and no filter nor scale factor is applied
(L2 data product). Mascon provides $\Delta TWS$ at equal-area $3^\circ$ spherical cap grid globally. In

contrast to the ITSG-DDK5 and GRGS solutions, the mascon uses a gain factor derived from the land surface model (LSM) to restore mitigated signals and reduce leakage errors (L3 data products) (Watkins et al., 2015; Wiese et al., 2016). Additionally, mascon provides the $\Delta TWS$ uncertainty together with the solution. The uncertainty is computed based on several geophysical models (see Watkins et al. (2015) and Wiese et al. (2016) for more details). The uncertainty information is not available in the ITSG-DDK5 or GRGS product.

The GRACE data are obtained between January 2003 and March 2016. After retrieval, the long-term mean value between January 2003 and March 2016 is computed and subtracted from the monthly products. To be consistent with CABLE grid spacing (see Sect. 4), the $\Delta TWS$ is computed using 0.5º spatial resolution. The coarse scale datasets (e.g., mascon, GRGS) are resampled to 0.5º×0.5º using the nearest grid values.

In this study, the independent GRACE solutions are used for two main reasons:

1. To obtain the $\Delta TWS$ values outside Australia. As shown in Eq. (7), the $\widehat{h}_o$ vector needs to be known, which can be from the GRACE-derived $\Delta TWS$ solution. We use the GRGS solutions as the GRGS solution is not subject to the filter choice and it provides $\Delta TWS$ at a spatial resolution comparable to the normal equation data.
2. To compare with the $\Delta TWS$ estimates from our approaches. All solutions are used to compare and validate our $\Delta TWS$ estimates.

**4. Hydrology model and validation data**

**4.1 Model setup**

The extensive description of the CABLE model is given in Decker (2015) and Ukkola et al. (2016). This section describes the model setup and specific changes applied to this study. CABLE is a public available land surface model and can be used to estimate soil moisture and groundwater in terms of volumetric water content every 3 hours at a 0.5º×0.5º spatial resolution. The soil moisture and groundwater storage can be simply computed by multiplying the estimates with thicknesses of various layers. For soil moisture, the thickness of 6 soil layers is 0.022, 0.058, 0.154, 0.409, 1.085, and 2.872 m, from top to bottom, respectively. The thickness of the groundwater layer is modeled to be 20 m uniformly. Recalling Eq. (3), $\Delta SM_{top}$ is defined as the soil moisture storage variation at the top 0.022 m thick layer, while $\Delta SM_{rz}$ is the variation accumulated over the second to the bottom soil layers (depth between 0.022 m and 4.6 m).

CABLE is initially forced with the data from the Global Soil Wetness Project Phase 3 (GSWP3), which is currently available until December 2010 (http://hydro.iis.u-tokyo.ac.jp/GSWP3, https://doi.org/10.20783/dias.501). We replace GSWP3 forcing data with GLDAS data (Rodell et al., 2004) to compute the water storage changes to 2016. The forcing data used in CABLE are precipitation, air temperature, snowfall rate, wind speed, humidity, surface pressure, and short-wave and long-wave downward radiations. To investigate the impact of different forcing data, the offline sensitivity study is conducted by comparing the water storage estimates computed using:

1. All 8 forcing data components of GSWP3,

2. GSWP3 data with replacing one component obtained from GLDAS forcing data.
It is found that the water storage estimate is most sensitive to the replacement of precipitation
data, as expected, and relatively less sensitive to the change of other forcing components. We
use the GLDAS forcing data in this study and also further test 7 different precipitation data
products (see more details in Sect. 4.2). The forcing data are up/down sampled to a 0.5º×0.5º
spatial grid to reconcile with the CABLE spatial resolution.

**4.2 Model uncertainty**
In this study, the CABLE uncertainty is derived from 210 ensemble estimates associated with
different forcing data and model parameters. The 7 different precipitation products (see Table
1) are used to run the model independently. Most products are available to present day while
GSWP3, Princeton, and MERRA are only available until December 2010, December 2012,
and February 2016, respectively. For each precipitation forcing, 30 ensembles are generated
by perturbing the model parameters within +/– 10% of the nominal values. The perturbed size
of 10% is similar to Dumedah and Walker (2014). Based on the CABLE structure, the $\Delta SM$
and $\Delta GWS$ estimates are most sensitive to the model parameters listed in Table 2. For
example, the fractions of clay, sand, and silt ($f_{clay}$, $f_{sand}$, $f_{silt}$) are used to compute soil
parameters including field capacity, hydraulic conductivity, and soil saturation which mainly
affect soil moisture storage. Similarly, the drainage parameters (e.g., $q_{sub}$, $f_p$) control the
amount of subsurface runoff, which has a direct impact on root zone soil moisture and
groundwater storages.
From ensemble generations, total $K = 210$ sets of the ensemble water storage estimates ($h_e$)
are obtained:
$$\mathcal{H}_R = [h_e|_{k=1} \quad h_e|_{k=2} \quad h_e|_{k=3} \quad ... \quad h_e|_{k=K}] \qquad (15)$$
and the mean value of $\mathcal{H}_R$ is computed as follows:
$$\tilde{h}_R = \frac{1}{K}\sum_{k=1}^{K}h_e|_k \qquad (16)$$
Note that due to the absence of GSWP3, Princeton, and MERRA data, the number of
ensembles reduces to $K = 180$ after December 2010, $K = 150$ after December 2012, and $K = $
120 after February 2016, respectively. The GC approach assumes that model errors are
normally distributed with zero mean. Any violation of this assumption will yield a bias in the
combined solutions. Therefore, the mean value is removed from each ensemble member,
$\mathcal{H}_R' = \mathcal{H}_R - \tilde{h}_R$, and the error covariance matrix of the model is empirically computed as:
$$\mathbf{C}_R = \mathcal{H}_R'(\mathcal{H}_R')^T/(K-1) \qquad (17)$$
The $\tilde{h}_R$ (Eq. (16)) and $\mathbf{C}_R$ (Eq. (17)) terms can be directly used in Eq. (9). The distribution of
model errors is demonstrated in Fig. 2. The figure illustrates the histogram of model errors
($\mathcal{H}_R'$) computed using 210 ensemble members of the model estimated $\Delta SM$ and $\Delta GWS$ in
Jan 2003. The histogram indicates that the model error may be approximately described by a
normal distribution as introduced in Eq. (1).
Furthermore, in practice, the sampling error caused by finite sample size might lead to
spurious correlations in the model covariance matrix (Hamill et al., 2001). The effect can be
reduced by applying an exponential decay with a particular spatial correlation length to $\mathbf{C_R}$. In
this study, the correlation length is determined based on the empirical covariance of model
estimated $\Delta TWS$. The covariance function of $\Delta TWS$ is firstly assumed isotropic, and it is
computed empirically based on the method given in Tscherning and Rapp (1974). The
distance where the maximum value of the variance decreases to half is defined as the
correlation length. The obtained values vary month-to-month, and the mean value of 250 km
is used in this study.
It is emphasized that the model omission error caused by imperfect modeling of hydrological
process within the LSM is not taken into account in the above description. The omission error
may increase the model covariance and introduce a bias as well. We account for the omission
error by increasing 20% of the model covariance. (i.e., multiplying $\mathbf{C_R}$ by 1.2). We determine
such omission error based on trial-and-error such that it increases the model error (due to the
omission error) but not exceeds the model error value reported by Dumedah and Walker
(2014). We acknowledge that this is only a simple practical way of accounting for the
omission error into the total model error.

## 4.3 Validation data

### 4.3.1 Satellite soil moisture observation

The satellite observed surface soil moisture data is obtained from the Advanced Microwave
Scanning Radiometer-Earth Observing System (AMSR-E) using the Land Parameter
Retrieval Model (Njoku et al., 2003). The observation is used to validate our estimates of top
soil moisture changes ($\Delta SM_{top}$). The AMSR-E product provides volumetric water content in
the top layer derived from a passive microwave data (from NASA EOS Aqua satellite) and
forward radiative transfer model. In this study, the level 3 product, available daily between
June 2002 and June 2011 at 0.25°×0.25° spatial resolution is used (Owe et al., 2008). The
measurements from ascending and descending overpasses are averaged for each frequency
band (C and X). Then, the monthly mean value is computed by averaging the daily data
within a month. To obtain the variation of the surface soil moisture, the long-term mean
between June 2002 and June 2011 is removed from the monthly data. Regarding the different
depth measured in CABLE and AMSR-E, the CDF-matching technique (Reichle and Koster,
2004) is used to reduce the bias between the top soil moisture model and the observation. The
CDF is built using the 2003-2004 data, and it is used for the entire period. There is no
satellite observed or ground measured root zone soil moisture data for meaningful
comparison with our results, particularly at continental scale. Validation of $\Delta SM_{rz}$ at regional
and continental scales is currently unachievable due to a complete lack of observations at this
spatial scale.

### 4.3.2 In-situ groundwater

The in-situ groundwater level from bore measurements are obtained from 2 different ground
observation networks (see Fig. 1). The data in Queensland are obtained from Department of
Natural Resources and Mines (DNRM) while the data in Victoria is from Department of
Environment and Primary Industries (DWPI). More than 10,000 measurements are available
from each network, but the data gap and outliers are present. Therefore, the bore

measurement is firstly filtered by removing the sites that present no data or data gap longer than 30 months during the study period.

To obtain the monthly mean value, the hourly or daily data are averaged in a particular month. The outliers are detected and fixed using the Hampel filter (Pearson, 2005) where the remaining data gaps are filled using the cubic spline interpolation. To obtain the groundwater level variation, the long-term mean groundwater level computed between the study period is removed from the monthly values. The groundwater level variation ($\Delta L$) is then converted to $\Delta GWS$ using $\Delta GWS = S_y \cdot \Delta L$, where $S_y$ is specific yield. Based on Chen et al. (2016), $S_y = 0.1$ is used for the Victoria network. Specific yields of Queensland's network have been found ranging from 0.045 (Rassam et al., 2013) to 0.06 (Welsh 2008), and an averaged $S_y = 0.05$ is used in this study. Finally, the mean value computed from all data (in each network) is used to represent the in-situ data of the network.

## 5. Results

### 5.1 Model-only performance

We study the model $\Delta TWS$ changes under different meteorological forcing and land parameterization. Total 210 estimates of monthly $TWS$ (sum of $SM_{top}$, $SM_{rz}$, and $GWS$) are obtained between January 2003 and March 2016 from the ensemble run based on 7 different precipitation inputs. Then, the averaged values of the $TWS$ estimates are computed from the 30 precipitation-associated ensemble members. This results in 7 sets of monthly mean $TWS$ estimates from 7 different precipitation data. For each set, the monthly $\Delta TWS$ is computed by removing the long-term mean computed between January 2003 and March 2016.

The precipitation-based $\Delta TWS$ are then compared with the GRACE-mascon solution (see Sect. 3.2) over 10 different Australian basins. The comparison is carried out between January 2003 and March 2016. Due to the availability of the data, the periods used are shorter in cases of GSWP3, Princeton, and MERRA precipitation (see Table 1). The metric used to evaluate a goodness of fit between CABLE run and GRACE mascon estimates is the Nash-Sutcliff (NS) coefficient (see Eq. (B1)) (Fig. 3).

Figure 3 demonstrates CABLE $\Delta TWS$ varies noticeably by precipitation as well as locations. The area-weighted average values (see Eq. (B2)) computed from Princeton, GSWP3, and TRMM yields the model $\Delta TWS$ reasonably agreeing with GRACE by giving the NS coefficient greater than 0.45, while MERRA, PERSIANN, and GLDAS show NS = ~0.3. The less agreement is mainly due to the quality of rainfall estimates over Australia. The NS of ECMWF is around 0.4.

All model ensembles are consistent with the GRACE data over the Timor Sea and inner parts of Australia (e.g., LKE, MRD, NWP) where the NS value can reach as high as 0.9 (see, e.g., TRMM over TIM). On the contrary, the less agreement is found mostly over the coastal basins. Very small or even negative NS values indicate the misfit between CABLE and GRACE mascon solutions, and they are observed over the Indian Ocean (see GLDAS), North East Coast (see GSWP3, PERSIANN, TRMM), South East Coast (see MERRA, TRMM), South West Coast (see GSWP3, GLDAS, MERRA), and South West Plateau (see MERRA).

By averaging all $\Delta TWS$ estimates from seven different precipitation datasets, the mean-
ensemble estimate (MN) delivers the best agreement with GRACE as seen by the highest
average NS value (MN of AVG = 0.55) among all ensembles. Particularly, NS values are
greater than 0.4 in all basins and no negative NS values are presented in MN. In average, it
can be clearly seen that using the mean value (MN) is a viable option to increase the overall
performance of the $\Delta TWS$ estimates. Therefore, only CABLE MN result will be used in
further analyses. The comparison with the GRGS GRACE solution was also evaluated (not
shown here) and the overall results are similar to Fig. 3.
**5.2 Impact of GRACE on storage estimates**
**5.2.1 Contribution of GRACE**
This section investigates the impact of the GC approach on the estimates of various water
storage components. The $\Delta TWS$ estimate obtained from the GC approach is demonstrated in
Sect. 5.1, by comparing with the independent GRACE mascon solution. Figure 3 shows the
GC result yields the highest NS values in all basins, outperforming all other CABLE runs. In
average (AVG), the NS value increases by ~35% (0.55 to 0.74) from the MN case. The
similar behaviour is also seen when compared with the GRGS GRACE solution (not shown);
the average NS value increases from 0.50 to 0.74. This is not surprising as the GC approach
uses the fundamental GRACE tracking data as GRACE mascon and GRGS solutions do.
Improvement of NS coefficient indicates merely the successfulness of integrating GRACE
data and the model estimates.
Figures 4 and 5 show the GC results of $\Delta TWS$ as well as $\Delta SM_{top}$, $\Delta SM_{rz}$, and $\Delta GWS$ in
different basins. The monthly time-series and the de-seasonalized time-series are shown. In
general, GRACE tends to increase $\Delta TWS$ when the model $\Delta TWS$ (MN) is predicted to be
underestimated (see e.g., LKE, MRD, NWP, SWP, TIM between 2011 and 2012) and by
decrease $\Delta TWS$ when determined to be overestimated (see all basins between 2008 and
2010). A clear example is seen over Gulf of Carpentaria (Fig. 4d), where CABLE
overestimates $\Delta TWS$ and produces phase delay between 2008 and 2010. The over estimated
amplitude and phase delay seen in CABLE $\Delta GWS$ during this above period (Fig. 4c) is
caused by an overestimation of soil and groundwater storage. The positively biased soil and
groundwater storage causes a phase delay by increasing the amount of time required for the
subsurface drainage (baseflow) to reduce to soil and groundwater stores. The overestimation
of water storage is the result of overestimated precipitation or underestimated
evapotranspiration. The amplitude and phase of the water storage estimate are adjusted
toward GRACE observation in the GC approach.
The impact of GRACE varies across the individual storage as well as across the geographical
location (climate regime). In general, the major contributors to $\Delta TWS$ are $\Delta SM_{rz}$ and $\Delta GWS$.
Due to a small store size (only ~2 cm thick), $\Delta SM_{top}$ contributes only ~2 % to $\Delta TWS$. As
such, $\Delta SM_{rz}$, and $\Delta GWS$ have greater variations, which commonly lead to greater uncertainty
compared to $\Delta SM_{top}$, and therefore, the stores anticipate greater shares from the GRACE
update. This behaviour is seen over all basins where the differences between CABLE-
simulated and GC $\Delta SM_{rz}$, and $\Delta GWS$ estimates are greater (compared to $\Delta SM_{top}$).
Furthermore, the impact of GRACE on $\Delta SM_{rz}$, and $\Delta GWS$ is different across the continent.
For example, over central and southern Australia (see e.g., LKE, MRD, NWP, SWP), the dry
climate is responsible for a small amount of groundwater recharge and most of the infiltration
is stored in soil compartments. In this climate condition, $\Delta SM_{rz}$ amplitude is significantly
larger than $\Delta GWS$ and it plays a greater role in $\Delta TWS$, and consequently, the GRACE
contribution is mostly seen in $\Delta SM_{rz}$ component. Different behaviour is seen over the
northern Australia (GOC, NEC, TIM) where $\Delta GWS$ amplitude are greater (~40 % of $\Delta TWS$)
compared to other basins (only ~17 % of $\Delta TWS$). This is due to the sufficient amount of
rainfall over the wet climate region, replenishing groundwater recharges and resulting in
greater variability in $\Delta GWS$. Therefore, compared to the dry climate basin, the GRACE
contributes to $\Delta GWS$ over these basins by the larger amount.

### 5.2.2 Impact on long-term trend estimates

The spatial patterns of the long-term trends of water storage changes over January 2003 and
March 2016 are analysed before and after applying the GC approach (Fig. 6). For
comparison, the long-term trends of $\Delta TWS$ derived from the ITSG-DDK5, mascon, and
GRGS solutions are shown in Fig. 7. From Fig. 6b, GRACE effectively changes the long-
term trend estimates in most basins in a way the spatial pattern of the $\Delta TWS$ trend of the GC
solution consistent to the GRACE solutions, while satisfying the model processes and
keeping the spatial resolution. The trend of $\Delta SM_{top}$ is insignificant (Fig. 6c) and the GC
approach does not change (Fig. 6d). The largest adjustment is seen in $\Delta SM_{rz}$ and $\Delta GWS$
components, to be consistent with the GRACE data in most basins (Fig. 6f, 6h).
GRACE shows significant changes in the $\Delta TWS$ trend estimates particularly over the
northern and western parts of the continent (Fig. 7). The model estimates around the Gulf of
Carpentaria basin show a strong negative trend that is inconsistent from the GRACE data. It
is found that underestimated precipitation after 2012 is likely the cause of such an
incompatible negative trend (see Fig. 4d). Applying the GC approach clearly improves the
trend (Fig. 6a vs. 6b). The other example is seen over the western part of the continent (see
rectangular area in Fig. 6a, 6b) where the averaged long-term trend of $\Delta TWS$ was predicted
to be –0.4 cm/year but changed to be –1.2 cm/year (see also Sect. 5.4) by the GC approach.
The precipitation over the western Australia is understood to be overestimated after 2012,
evidently seen by that the model $\Delta TWS$ is always greater than the GC solution (see e.g., Fig.
4h, 5d, 5p). The GC approach reveals that the water loss over the western Australia is at least
twice greater than what has predicted by the CABLE model.
In addition, the shortage of water storage in the south-eastern part of the continent from the
millennium drought (McGrath et al., 2012) has been recovered (seen as a positive water
storage trend in Fig. 6) after the rainfall between 2009 and 2012, while the western part is
still drying out (seen as negative trends). The trend estimates in terms of mass change are
discussed in more detail in Sect. 5.4.

### 5.2.3 Reduction of uncertainty

Influenced by climate pattern, the uncertainty of water storage estimates significantly varies
across Australia. The uncertainty of the model estimate is computed from the variability
induced by different precipitation and model parameters while the uncertainty of GC solution
is computed using Eq. (11). As expected, larger uncertainties are observed in $\Delta SM_{rz}$ and
$\Delta GWS$ than in $\Delta SM_{top}$ (an order of magnitude smaller) since $\Delta SM_{top}$ is smaller than others
(Fig. 8). Over the wet basins, larger amplitude of the water storage leads to larger uncertainty,
seen over Gulf of Carpentaria, North East Coast, South East Coast, and Timor Sea where the
CABLE-simulated $\Delta TWS$ uncertainty is approximately 28 % larger than other basins. The
smaller uncertainty is found over the dry regions (e.g., LKE, SWP). In most basins, the
uncertainty of $\Delta SM_{rz}$ is larger than the $\Delta GWS$, except the wet basins (e.g., GOC, NEC, TIM)
where the greater groundwater recharge leads to a larger uncertainty of $\Delta GWS$.
Figure 8 demonstrates how much the formal error of each of storage components is reduced
by the GC approach. Overall, the estimated CABLE uncertainties averaged over all basins
(AVG) are 0.2, 4.0, 4.0, and 5.7 cm for $\Delta SM_{top}$, $\Delta SM_{rz}$ , $\Delta GWS$, and $\Delta TWS$, respectively.
With the GC approach, the uncertainties of $\Delta SM_{top}$, $\Delta SM_{rz}$ , $\Delta GWS$, and $\Delta TWS$ decrease by
approximately 26%, 35%, 39%, and 37%, respectively.
It is worth mentioning that the model uncertainty is mainly influenced by the meteorological
forcing data. The uncertainty of precipitation derived from seven different precipitation
products is shown in Fig. 8e. The spatial pattern of the precipitation uncertainty is correlated
with the uncertainty of water storage estimates. The larger water storage uncertainty is
deduced from the larger precipitation uncertainty. The quality of precipitation forcing data is
found to be an important factor to determine the accuracy of water storage computation.

**5.3 Comparison with independent data**
**5.3.1 Soil moisture**
The $\Delta SM_{top}$ estimates are compared with the AMSR-E derived soil moisture. The processing
of AMSR-E data is described in Sect 4.3.1. The performance is assessed using Nash-Sutcliff
coefficients, given in Table 3. In general, CABLE (MN) shows a good performance in the top
soil moisture simulation showing NS value of >0.4 for most of the basins. The top soil
moisture estimate shows slightly better agreement with the C-band measurement of the
AMSR-E product. This is likely caused by the greater emitting depth of the C-band
measurement (~1 cm), which is closer to the depth of the top soil layer (~2 cm) used in this
study (Njoku et al., 2003).
The GC approach leads to a small bit of improvement of the top soil estimate consistently
from C- and X-band measurements and from all basins. No degradation of the NS value is
observed in the GC solutions. The largest improvement is seen over LKE and NEC, where
NS increases by 10 – 15%.  For other regions, the change in the NS coefficient may be
incremental.

**5.3.2 Groundwater**
The $\Delta GWS$ estimates from the model and the GC method are compared with the in situ data
obtained from 2 different ground networks in Queensland and Vitoria. For each network, all
$\Delta GWS$ data inside the groundwater network boundary (see polygons in Fig. 1) are used to
compute the average $\Delta GWS$ time series. From the comparison given in Fig. 9, it is found that
the GC solutions of $\Delta GWS$ follows the overall inter-annual pattern of CABLE but with a
considerably larger amplitude. This results in a better agreement with the in situ $\Delta GWS$ data
seen from both networks. The NS coefficient of $\Delta GWS$ between the estimates and the in situ
data are given in Table 4. The CABLE $\Delta GWS$ performs significantly better in Queensland
(NS = ~0.5) than Victoria (NS = ~0.3). Significant improvement is found from the GC
solutions in both networks, where the NS value increases from 0.5 to 0.6 (~ 22 %) in
Queensland and from 0.3 to 0.6 (~85 %) in Victoria. Even greater improvement is seen when
the inter-annual patterns are compared. The NS value increase from 0.5 to 0.7 (~ 32 %), and
0.4 to 0.8 (~93 %) in Queensland and Victoria, respectively.
The comparison of the long-term trend of $\Delta GWS$ is also evaluated. The estimated trends in
Queensland and Victoria are given in Table 4. Beneficially from the GC approach, the $\Delta GWS$
trend is improved by approximately 20 % (from 0.4 to 0.6, compared to 1.6 cm/year) in
Queensland. Increasing of $\Delta GWS$ is mainly influenced by the large amount of rainfall during
the 2009 – 2012 La Niña episodes (see Fig. 9a). In Victoria, significant improvement of
$\Delta GWS$ trend by about 76 % (from 0.1 to –0.2, compared to –0.3 cm/year) is observed.
Similar improvement of long-term trend estimates is seen in de-seasonalized time series
(improves by ~15 % in Queensland and by ~74 % in Victoria). Decreasing of $\Delta GWS$ in
Victoria is mainly due to the highly-demanded groundwater consumption by agriculture and
domestic activities (van Dijk et al., 2007; Chen et al., 2016). As the groundwater
consumption is not parameterized in CABLE, the decreasing of $\Delta GWS$ estimate cannot
properly captured in the model simulation. Applying GC approach effectively reduces the
model deficiency and improves the quality of the groundwater estimations.

## 5.4 Assessment of mass variation in the past 13 years

Australia experiences significant climate variability; for example, the millennium drought
starting from late '90 (Van Dijk et al., 2013) and extremely wet condition during several La
Niña episodes (Trenberth 2012; Han 2017). These periods are referred as "Big Dry" and "Big
Wet" (Ummenhofer et al., 2009; Xie et al., 2016). To understand the total water storage
(mass) variation influenced by these two distinct climate variabilities, the water storage
change obtained from the GC approach during Big Dry and Big Wet is separately
investigated over 10 basins. The time window between January 2003 and December 2009 is
defined as the Big Dry period while between January 2010 and December 2012 is defined as
the Big Wet period following Xie et al. (2016). In each period, the long-term trends of GC
estimates of $\Delta TWS$, $\Delta SM_{top}$, $\Delta SM_{rz}$ , and $\Delta GWS$ are firstly calculated. Then, the total water
storage variation (in meter) is simply obtained by multiplying the long-term trend (in m/year)
with the number of years in the specific period, 7 years for Big Dry and 3 years for Big Wet.
To obtain the mass variation, the water storage variation is multiplied by the area of the basin
and the density of water (1000 kg/m$^3$). The estimated mass variations during Big Dry and Big
Wet are displayed in Fig. 10. The long-term mass variation of the entire period (January 2003
– March 2016) is also shown.
During Big Dry (2003 – 2009), a significant loss of total storage (40 – 60 Gton over 7 years)
is observed over LKE, MRD, NWP, and SWP basins. The largest groundwater loss of >20
Gton is found from LKE and MRD. No significant change is observed over the tropical
climate regions (e.g., GOC, NEC). The mass loss mostly occurs in the root zone and

groundwater compartments where the sum of $\Delta SM_{rz}$ and $\Delta GWS$ explains more than 90% of the $\Delta TWS$ value. The mass loss is also observed in $\Delta SM_{top}$ but >10 times smaller than $\Delta SM_{rz}$ and $\Delta GWS$.

During Big Wet (2010 – 2012), the basins like LKE, MRD and TIM exhibit the significant total storage gain of >100 Gton. The gain is particularly larger in $\Delta SM_{rz}$ over the basins that experienced the significant loss during Big Dry. For example, over LKE and MRD, the gain of $\Delta SM_{rz}$ is approximately 2 – 3 times greater than $\Delta GWS$. It implies that most of the infiltration (from the 2009 – 2012 La Niña rainfall) is stored as soil moisture through the long drought period, and that the groundwater recharge is secondary to the $\Delta SM_{rz}$ increase.

The opposite behaviour is observed over the basins (such as NEC and GOC) that experienced mass gain during Big Dry. The water storage gain is greater in $\Delta GWS$ compared to $\Delta SM_{rz}$. In NEC, $\Delta GWS$ gain is ~8 times larger than $\Delta SM_{rz}$ during Big Wet. The soil compartment may be saturated during Big Dry and additional infiltration from the Big Wet precipitation leads to an increased groundwater recharge. The $\Delta SM_{rz}$ loss observed over GOC is simply caused by the timing selection of Big Wet period, which ends earlier (~2011) in GOC than in other basins. The $\Delta SM_{rz}$ gain becomes ~26 Gton if the Big Wet period is defined as 2008 – 2011. During the post-Big Wet period (2012 and afterwards), the decreasing trend of water storage is observed from all basins (see Fig. 4, 5). This is mainly caused by the decrease in precipitation after 2012 and by gradual water loss through evapotranspiration (Fasullo et al., 2013).

The overall water storage change in the last 13 years demonstrates that the severe water loss from most basins during Big Dry (the millennium drought) is balanced with the gain during Big Wet (the La Niña). The negative $\Delta TWS$ estimated during Big Dry becomes positive in LKE, MRD, and SEC and less negative in TIM, and the greatest gain is observed from NEC by ~50 Gton during 13 year-period (see Fig. 10c). However, the water mass loss is still detected over the western basins (e.g., IND, NWP, SWP, SWC), and their magnitudes are even larger than the mass loss during Big Dry. For example, the greatest $\Delta TWS$ loss of ~79 Gton is observed over NWP, which is ~25 Gton greater than the loss during Big Dry (see Fig. 10a and 10c). The basin is less affected by the La Niña, and the rainfall during Big Wet is clearly inadequate to support the water storage recovery in the basin. Rainfall deficiency also reduces the groundwater recharge, resulting in even more decreasing of $\Delta GWS$, compared to the Millennium Drought period (see Fig. 10j and 10l). The continual decrease in water storage over western basins is likely caused by the interaction of complex climate patterns like El Niño Southern Oscillation, Indian Ocean Dipole, and Southern Annular Mode cycles (Australian Bureau of Meteorology, 2012; Xie et al., 2016).

## 5.5 Comparison of GC approach with alternatives

The simplest approach to estimate $\Delta GWS$ is to subtract the model soil moisture component from GRACE $\Delta TWS$ data, without considering uncertainty in the model output, as used in Rodell et al. (2009) and Famiglietti et al. (2011). This method is called Approach 1 (App 1). In Approach 2 (App 2) as in Tangdamrongsub et al. (2017), by accounting for the uncertainty of model outputs and GRACE data, the water storage states are updated through a Kalman filter:

$$\widehat{h}_R = \widetilde{h}_R + \mathbf{H}\mathbf{C}_R^T(\mathbf{H}\mathbf{R}\mathbf{H}^T + \mathbf{C}_R)^{-1}(b - \mathbf{H}\widetilde{h}_R) \tag{18}$$

where $\widetilde{h}_R$, $\mathbf{H}$, $\mathbf{C}_R$ are described in Sect. 2, $b$ is an observation vector containing GRACE-derived $\Delta TWS$, and $\mathbf{R}$ is an error variance-covariance matrix of the observation. The GRACE-derived $\Delta TWS$ and its error information is obtained from the mascon solution. The matrix $\mathbf{R}$ is a (diagonal) error variance matrix since no covariance information is given in the mascon product. Note that the model uncertainty remains the same as in GC approach (Sect. 4.2). The different results from App1 and App2 are mainly attributed to the different estimates of the uncertainty.

The $\Delta GWS$ estimates from App1, App2 and GC in Queensland and Victoria are shown in Fig. 11. It is clearly seen that $\Delta GWS$ from App1 are overestimated while the one from App2 fits the ground data significantly better. This behaviour was also seen in Tangdamrongsub et al. (2017) that the water storage estimates tend to be overestimated when error components such as spatial correlation error were neglected as in App1. $\Delta GWS$ from App2 shows clear improvements in terms of NS coefficients in both networks. Considering the de-seasonalized $\Delta GWS$ estimates, in Queensland, the trend increases from 0.39 ± 0.03 to 0.42 ± 0.03 cm/year (improves by 1.5%), and the NS value increases from 0.46 to 0.53. In Victoria, the trend decreases from 0.73 ± 0.10 to 0.46 ± 0.05 cm/year (improves by 27%), and the NS value increases from –0.89 to 0.30. Although App2 is not yet as good as the GC solution based on the most comprehensive error propagation, this simple test demonstrates an important of considering the uncertainty. The reason of App2 being less accurate than GC is likely due to too simplified error information implemented in App2.

## 6. Conclusion

This study presents an approach of combining the raw GRACE observation with model simulation to improve water storage estimates over Australia. Distinct from other methods, we exploit the fundamental GRACE satellite tracking data and the full data error variance-covariance information to avoid alteration of signal and measurement error information present in higher level data products.

We compare groundwater storage estimates from GC approach and two other approaches, subject to inclusion of GRACE uncertainty in the $\Delta GWS$ calculation. Validating three results of $\Delta GWS$ against the in situ groundwater data, we find that the GC approach delivers the most accurate groundwater estimate, followed by the approach based on incomplete information of GRACE's data error. The poorest estimate of groundwater storage is seen when the GRACE uncertainty is completely ignored. This confirms the critical value of using the complete GRACE signal and error information at the raw data level.

The analysis of water storage change between 2003 and 2016 reveals that half of the continent (5 out of 10 basins) is still not fully recovered from the Millennium Drought. The TWS decrease in Western Australia has been most characteristic, and the GC approach finds that the water loss mainly occurs in groundwater layer. Rainfall inadequacy is attributed to the continual dry condition, leading to a greater decreasing of groundwater recharge and storage over Western Australia.

The land surface model we used is deficient in anthropogenic groundwater consumption. The
model calibration will never help, and the groundwater consumption must be brought in by
external sources. On the contrary, the statistical approach like our GC approach may be
useful to fill in the missing component and lead to a more comprehensive water storage
inventory.
However, it is difficult to constrain different water storage components by only using total
storage observation like GRACE. In addition, it is challenging to improve surface soil
moisture varying rapidly in time, using a monthly mean GRACE observation. Tian et al.
(2017) utilized the satellite soil moisture observation from the Soil Moisture and Ocean
Salinity (SMOS, Kerr et al., 2001) in addition to GRACE data for their data assimilation and
showed a clear improvement in the top soil moisture estimate. The GC approach with
complementary observations at higher temporal resolution should be considered particularly
to enhance the surface soil moisture computation.
Furthermore, the GC approach can be simply extended for GRACE data assimilation.
Assimilating the raw GRACE data into land surface models like CABLE enables the model
state and parameter to be adjusted with the realistic error information, allowing more reliable
storage computation.

## Acknowledgement

This work is funded by The University of Newcastle to support NASA's GRACE and
GRACE Follow-On projects as an international science team member to the missions. MD
was supported by ARC Centre of Excellence for Climate Systems Science. HK was
supported by Japan Society for the Promotion of Science KAKENHI (16H06291).  We thank
Torsten Mayer-Gürr for GRACE data products in the form of the least-squares normal
equations. We also thank three anonymous reviewers for helping us improve the manuscript.

**Appendix A: Least-squares normal equation of GRACE**

A linearized GRACE satellite-tracking observation equation is formulated as:

$$\boldsymbol{y} = \mathbf{A}\boldsymbol{x} + \boldsymbol{e}; \boldsymbol{e} \sim \mathcal{N}(\mathbf{0}, \boldsymbol{\Sigma}), \qquad (A1)$$

where $\boldsymbol{y}$ is the observation vector containing various kinds of L1B data including the inter-satellite ranging data, $\mathbf{A}$ is the design (partial derivative) matrix relating the data and the Earth gravity field variations, $\boldsymbol{x}$ contains the Stokes coefficients of time-varying geopotential fields (e.g., Wahr et al., 1998), and $\boldsymbol{e}$ is the L1B data noise, which has zero mean and covariance $\boldsymbol{\Sigma}$. Eq. (A1) can be modified explicitly in terms of soil moisture and groundwater storage variations as:

$$\boldsymbol{y} = \mathbf{AS}\overline{\mathbf{Y}}\mathbf{H}\boldsymbol{h} + \boldsymbol{e}; \boldsymbol{e} \sim \mathcal{N}(\mathbf{0}, \boldsymbol{\Sigma}), \qquad (A2)$$

where $\mathbf{S}$ contains a factor used to convert $\Delta TWS$ to geopotential coefficients considering the load Love numbers (e.g., Wahr et al., 1998), $\overline{\mathbf{Y}}$ converts the gridded data into the corresponding spherical harmonic coefficients. For convenience, the term $\mathbf{Y} = \mathbf{S}\overline{\mathbf{Y}}$ is used in the further derivation. A least-squares solution of Eq. (A2) is given as:

$$(\mathbf{H}^T\mathbf{Y}^T\mathbf{A}^T\boldsymbol{\Sigma}^{-1}\mathbf{AYH})\widehat{\boldsymbol{h}} = \mathbf{H}^T\mathbf{Y}^T\mathbf{A}^T\boldsymbol{\Sigma}^{-1}\boldsymbol{y}. \qquad (A3)$$

It can be simplified as:

$$\mathbf{H}^T\mathbf{Y}^T\mathbf{N}\,\mathbf{YH}\,\widehat{\boldsymbol{h}} = \mathbf{H}^T\mathbf{Y}^T\boldsymbol{c}, \qquad (A4)$$

where $\mathbf{N} = \mathbf{A}^T\boldsymbol{\Sigma}^{-1}\mathbf{A}$ and $\boldsymbol{c} = \mathbf{A}^T\boldsymbol{\Sigma}^{-1}\boldsymbol{y}$. Eq. (A4) is identical to Eq. (5).

**Appendix B: Nash-Sutcliff coefficient and area-weighted average**

Nash-Sutcliff coefficient (NS) is computed as follows:

$$NS = 1 - \frac{\sum_{i=1}^{N}(\boldsymbol{y_i} - \widehat{\boldsymbol{x_i}})^2}{\sum_{i=1}^{N}(\boldsymbol{y_i} - \overline{\boldsymbol{y}})^2} \qquad (B1)$$

where $\boldsymbol{y}$ is an observation vector, $\overline{\boldsymbol{y}}$ is the mean of the observation, $\widehat{\boldsymbol{x}}$ is a vector containing the simulated result, $i$ is the index of observation, and $N$ is the number of observation.

Area-weighted average ($\bar{Z}$) is compute as follows:

$$\bar{Z} = \frac{\sum_{j=1}^{M} w_j \bar{z}_j}{\sum_{j=1}^{M} w_j} \qquad (B2)$$

where $w$ is the area size, $\bar{z}$ is the mean value inside the considered area, $j$ is the area index, and $M$ is the number of considered area.

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

**Table 1.** Precipitation data from 7 different products used in this study, the Global Soil
Wetness Project Phase 3 (GSWP3), the Global Land Data Assimilation System (GLDAS),
the Tropical Rainfall Measuring Mission (TRMM), the Modern-Era Retrospective Analysis
for Research and Applications (MERRA), the European Centre for Medium-Range Weather
Forecasts (ECMWF), the Princeton's Global Meteorological Forcing Dataset (Princeton), and
the Precipitation Estimation from Remotely Sensed Information using Artificial Neural
Networks (PERSIANN). The temporal resolution of all products is 3 hours. Most products
are available to present while GSWP3, MERRA, and Princeton terminate earlier.

| Product | Availability | Spatial resolution | References |
|---|---|---|---|
| GSWP3 | 1901/01 – 2010/12 | 0.5°×0.5° | http://hydro.iis.u-tokyo.ac.jp/GSWP3 |
| GLDAS (NOAH025SUBP 3H) | 2000/03 – present | 0.25°×0.25° | Rodell et al. (2004) |
| TRMM (3B42) | 1998/01 – present | 0.25°×0.25° | Huffman et al. (2007) |
| MERRA (MSTMNXMLD.5.2.0) | 1980/01 – 2016/02 | 0.5°×0.67° | Rienecker et al. (2011) |
| ECMWF (ERA-Interim) | 1979/01 – present | 0.75°×0.75° | Dee et al. (2011) |
| Princeton (V2 0.5°) | 1987/01 – 2012/12 | 0.5°×0.5° | Sheffield et al. (2005) |
| PERSIANN (3 hr) | 2002/03 – present | 0.25°×0.25° | Sorooshian et al. (2000) |


**Table 2.** Model parameters that are sensitive to SM and GWS estimates. The following
parameters were perturbed using the additive noise with the boundary conditions given in the
last column. The further parameter description can be found in Decker (2015) and Ukkola et
al. (2016).

| Parameter | Name | Spatial variability | Perturbed range |
|---|---|---|---|
| $f_{clay}, f_{sand}, f_{silt}$ | Fraction of clay, sand, and silt | Yes | 0 – 1 |
| $f_{sat}$ | Fraction of grid cell that is saturated | No | 810 – 990 |
| $q_{sub}$ | Maximum rate of subsurface drainage assuming a fully saturated soil column | No | 0.009 – 0.01 |
| $f_P$ | Tuneable parameter controlling drainage speed | No | 1.9 – 2.2 |


**Table 3**. NS coefficients between top soil moisture estimates and the satellite soil moisture observations from AMSR-E products over 10 different Australian basins. The area-weighted average value (AVG) is also shown.

|  | C-band | | X-band | |
|---|---|---|---|---|
|  | CABLE | GC | CABLE | GC |
| GOC | 0.67 | 0.68 | 0.58 | 0.60 |
| IND | 0.53 | 0.54 | 0.41 | 0.41 |
| LKE | 0.48 | 0.53 | 0.36 | 0.42 |
| MRD | 0.77 | 0.80 | 0.75 | 0.78 |
| NEC | 0.34 | 0.39 | 0.14 | 0.19 |
| NWP | 0.33 | 0.36 | 0.38 | 0.42 |
| SEC | 0.68 | 0.68 | 0.69 | 0.71 |
| SWC | 0.85 | 0.85 | 0.89 | 0.89 |
| SWP | 0.55 | 0.56 | 0.46 | 0.48 |
| TIM | 0.44 | 0.45 | 0.16 | 0.16 |
| AVG | 0.53 | 0.56 | 0.47 | 0.50 |

**Table 4**. NS coefficient and long-term trend of $\Delta GWS$ estimated from the model-only and GC solutions in Queensland and Victoria groundwater network. The long-term trend of the in-situ data is also shown.

|  | Queensland | | | Victoria | | |
|---|---|---|---|---|---|---|
|  | In-situ | CABLE | GC | In-situ | CABLE | GC |
| **Original time-series** | | | | | | |
| NS [-] | - | 0.49 | 0.60 | - | 0.34 | 0.63 |
| Trend [cm/year] | $1.60 \pm 0.05$ | $0.39 \pm 0.02$ | $0.63 \pm 0.05$ | $-0.27 \pm 0.05$ | $0.10 \pm 002$ | $-0.18 \pm 0.03$ |
| **De-seasonalized time-series** | | | | | | |
| NS [-] | - | 0.50 | 0.66 | - | 0.43 | 0.83 |
| Trend [cm/year] | $1.60 \pm 0.05$ | $0.39 \pm 0.02$ | $0.57 \pm 0.04$ | $-0.25 \pm 0.05$ | $0.10 \pm 0.02$ | $-0.16 \pm 0.03$ |

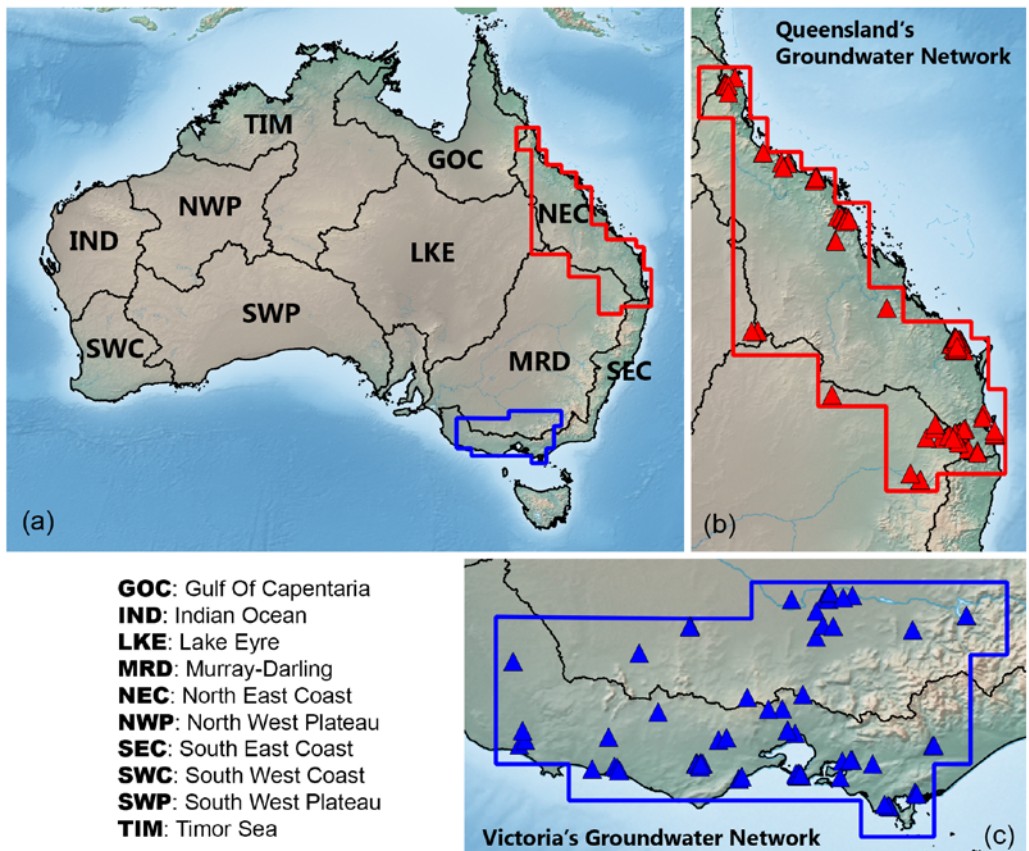


**Figure 1.** (a) Geographical location of 10 Australian river basins. Red and blue polygons indicate the boundaries of groundwater networks in Queensland (b) and Victoria (c), respectively. Triangles (in b and c) represent the selected bore locations used in this study.


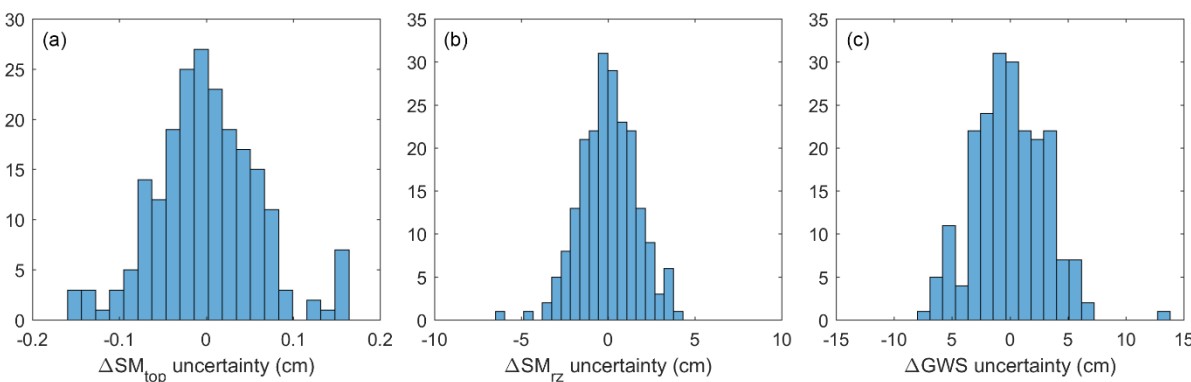


**Figure 2.** Histograms of the model errors computed from 210 ensemble members ($\mathcal{H}_R{}'$)
without the mean. The basin averaged values (from all 10 Australian basins) of January 2003,
for example, are shown.

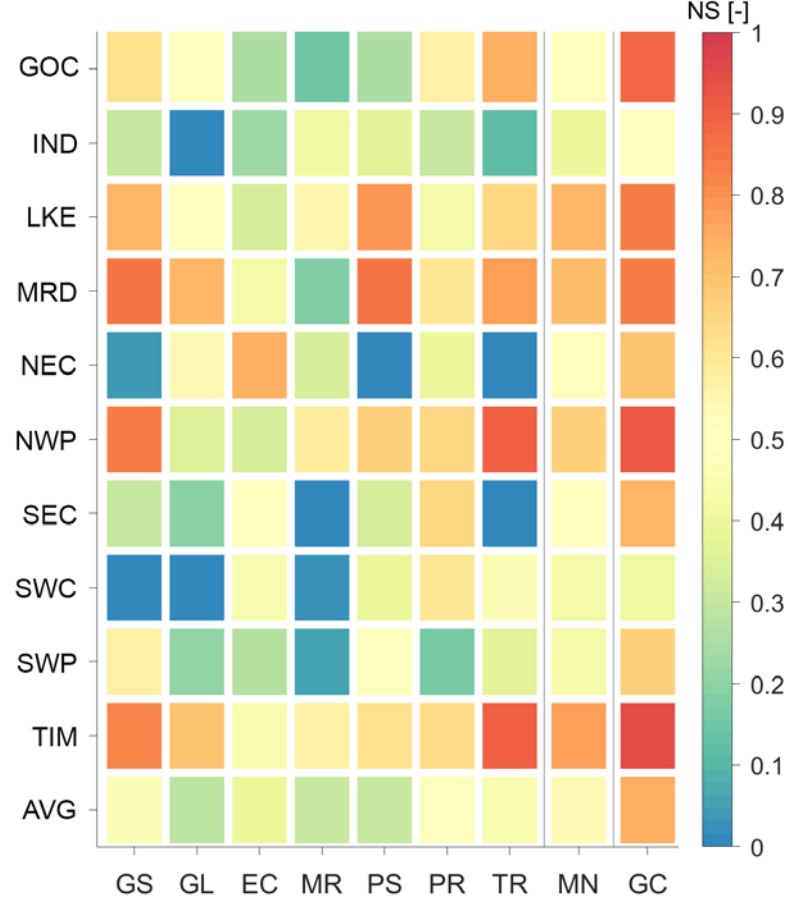


**Figure 3.** NS coefficients between the model and GRACE-mascon $\Delta TWS$ over 10 Australian basins (in ordinate). The NS values were computed based on CABLE $\Delta TWS$ computed with 7 different precipitation data (in abscissa), GSWP3 (GS), GLDAS (GL), ECMWF (EC), MERRA (MR), PERSIANN (PR), TRMM (TR). The NS value of the mean $\Delta TWS$ estimates (the average of 7 variants) is also shown (MN). The area-weighted average NS value over all basins is also shown (AVG). The NS value of $\Delta TWS$ from the GRACE-combined (GC) approach is shown in the last column. The full name of the basins can be found in Fig. 1.

926

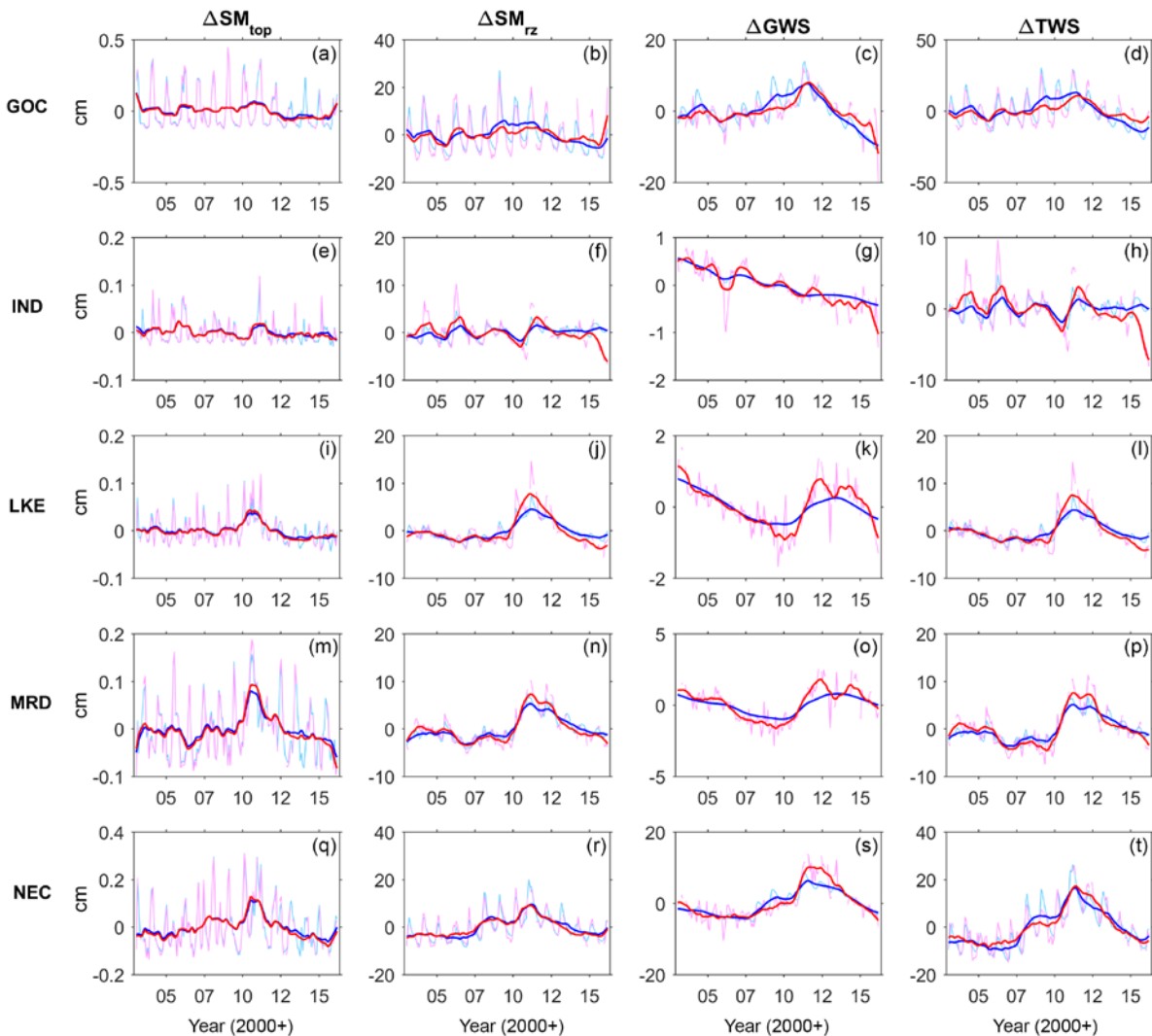

927

**Figure 4.** The monthly time series of $\Delta SM_{top}$, $\Delta SM_{rz}$, $\Delta GWS$, and $\Delta TWS$ estimated from
model (blue) and GC (red) solutions over Gulf of Carpentaria (GOC), Indian Ocean (IND),
Lake Eyre (LKE), Murray-Darling (MRD), and North East Coast (NEC). The de-
seasonalized time series is also shown.

932

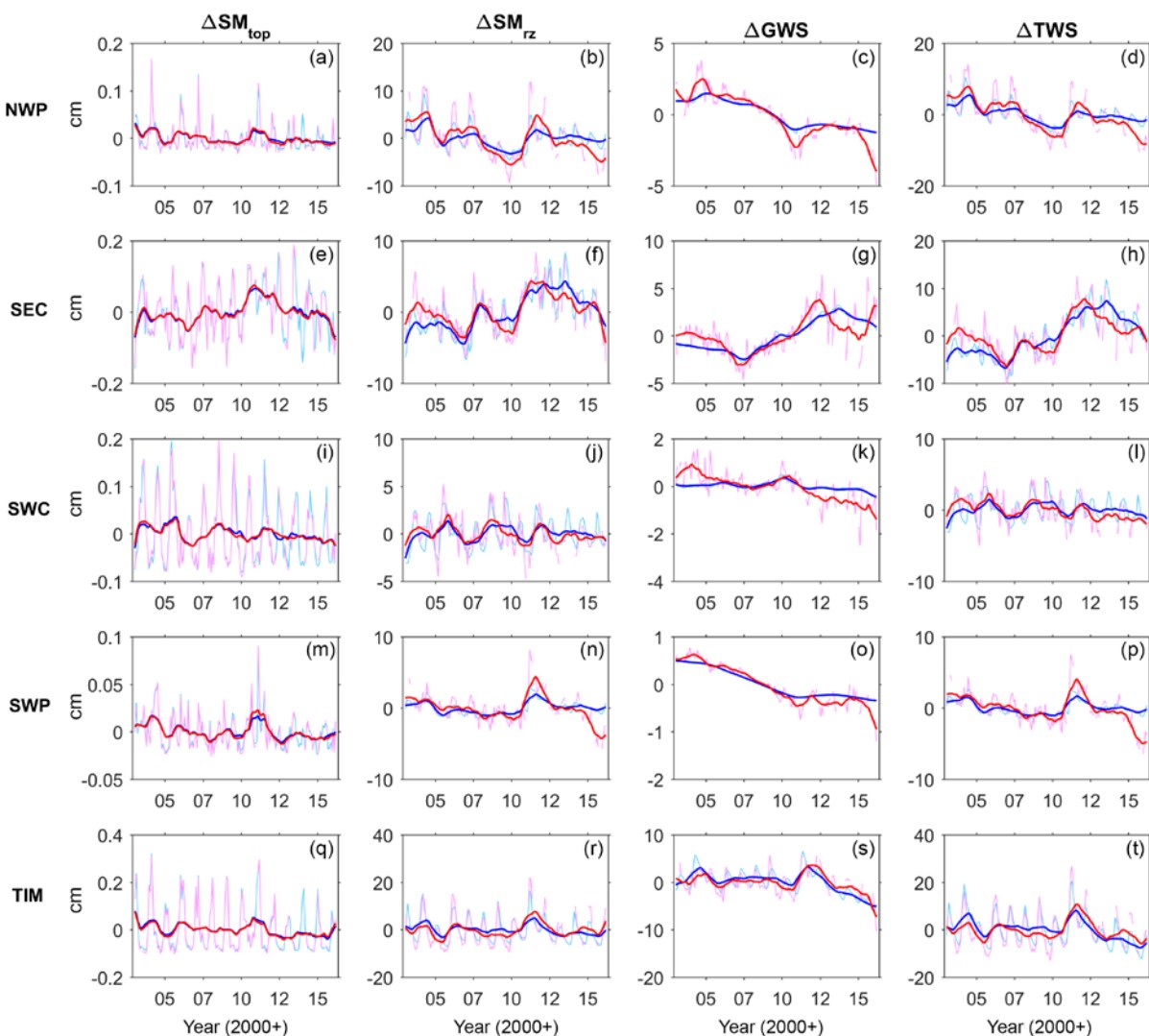

933

**Figure 5.** Similar to Fig. 3, but estimated over North West Plateau (NWP), South East Coast
(SEC), South West Coast (SWC), South West Plateau (SWP), and Timor Sea (TIM).

936

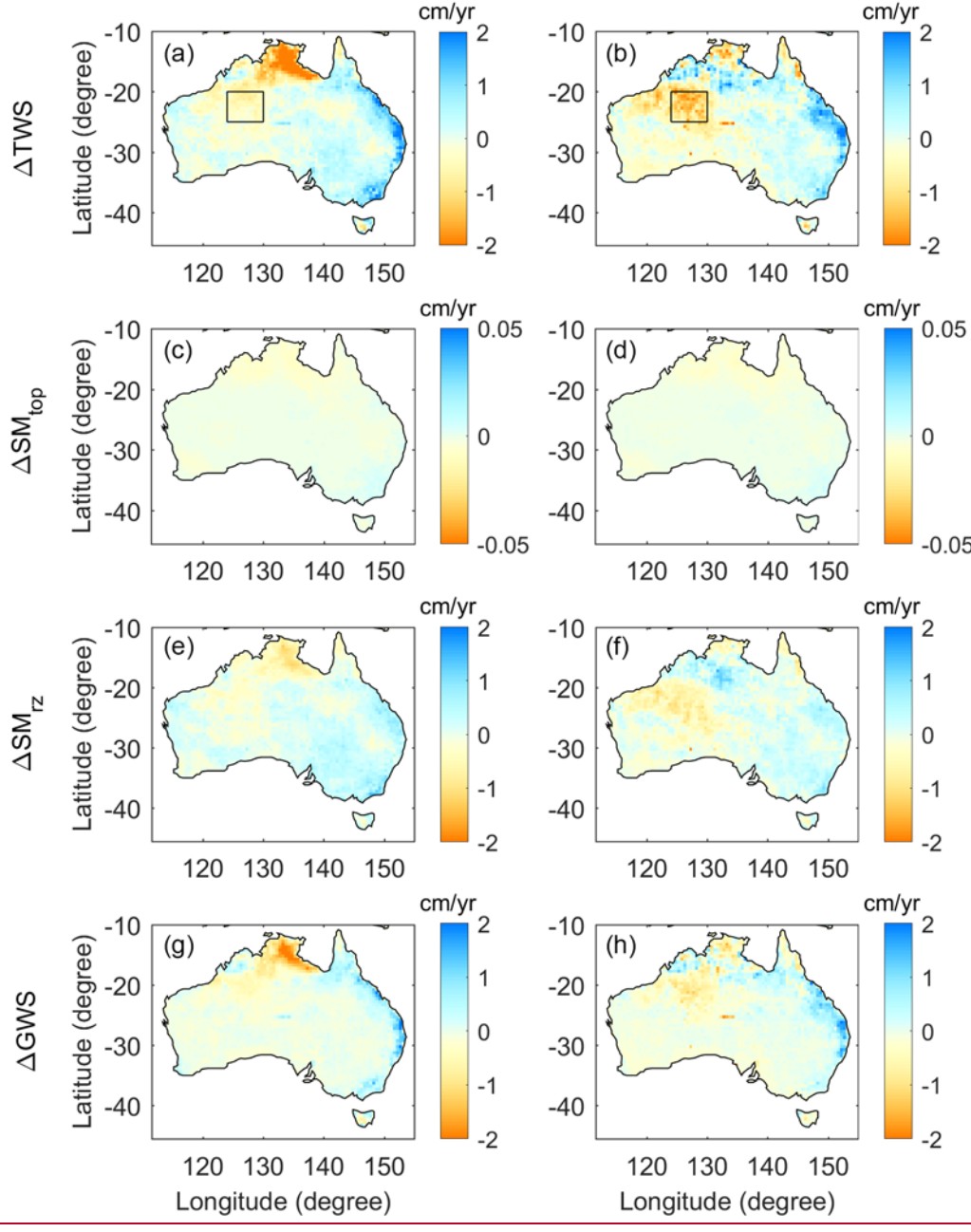

**Figure 6.** Long-term trends of $\Delta TWS$ (a, b), $\Delta SM_{top}$ (c, d), $\Delta SM_{rz}$ (e, f), and $\Delta GWS$ (g, h) estimated from the model-only (left) and the GC solutions (right). The eastern part of North West Plateau basin is shown as a rectangle polygon in (a) and (b).

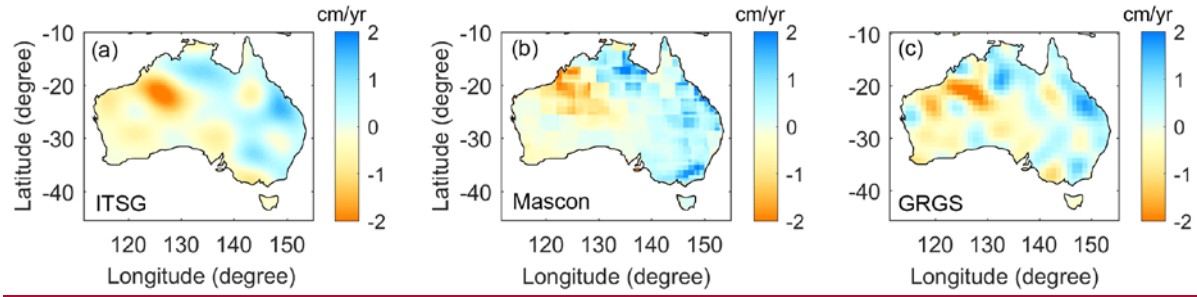


**Figure 7.** Long-term trends of GRACE-derived $\Delta TWS$ from ITSG-DDK5 (a), mascon (b), and GRGS solution (c).


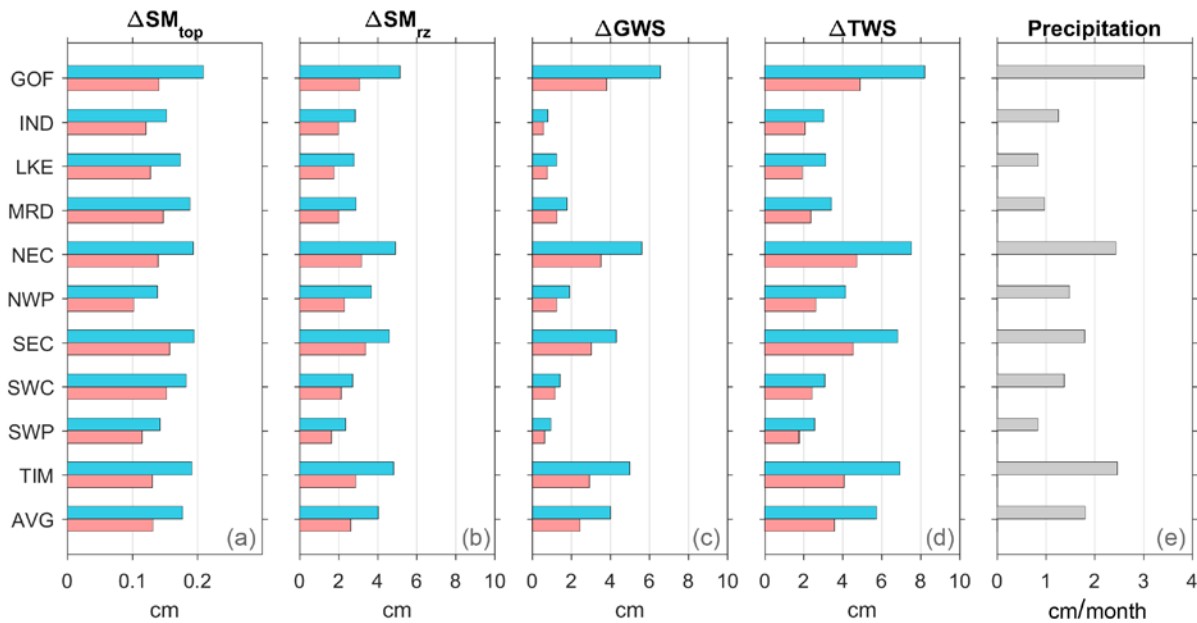


**Figure 8.** Uncertainties of $\Delta SM_{top}$, $\Delta SM_{rz}$, $\Delta GWS$, and $\Delta TWS$ estimated from the model (blue) and the GC solutions (red) in 10 different Australian basins. The uncertainty of the precipitation is shown in (e). The area-weighted average value (AVG) is also shown.


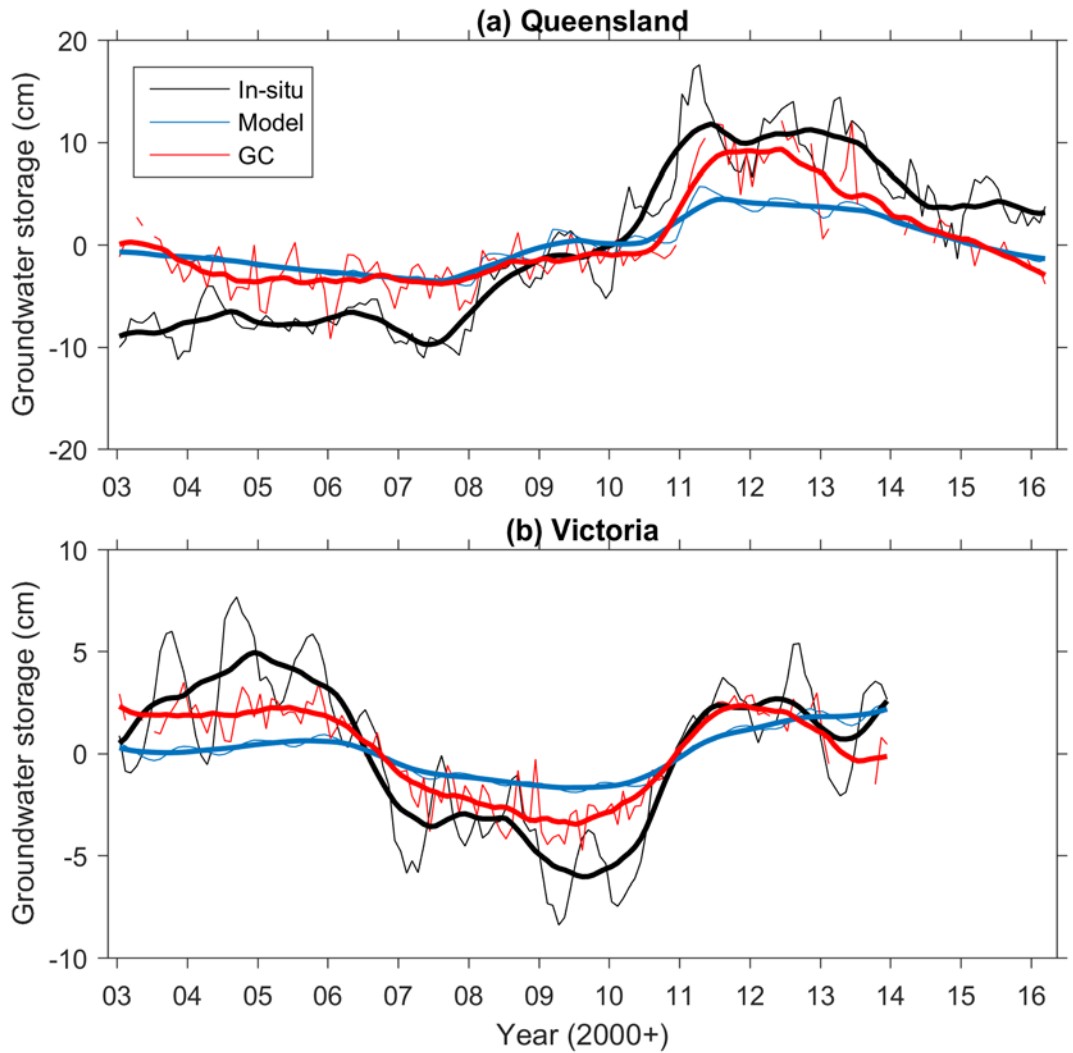


**Figure 9.** The monthly time series of $\Delta GWS$ estimated from the model, GC solutions, and
measured from the in situ groundwater network in Queensland (a) and Victoria (b). De-
seasonalized time series are shown in thick lines.


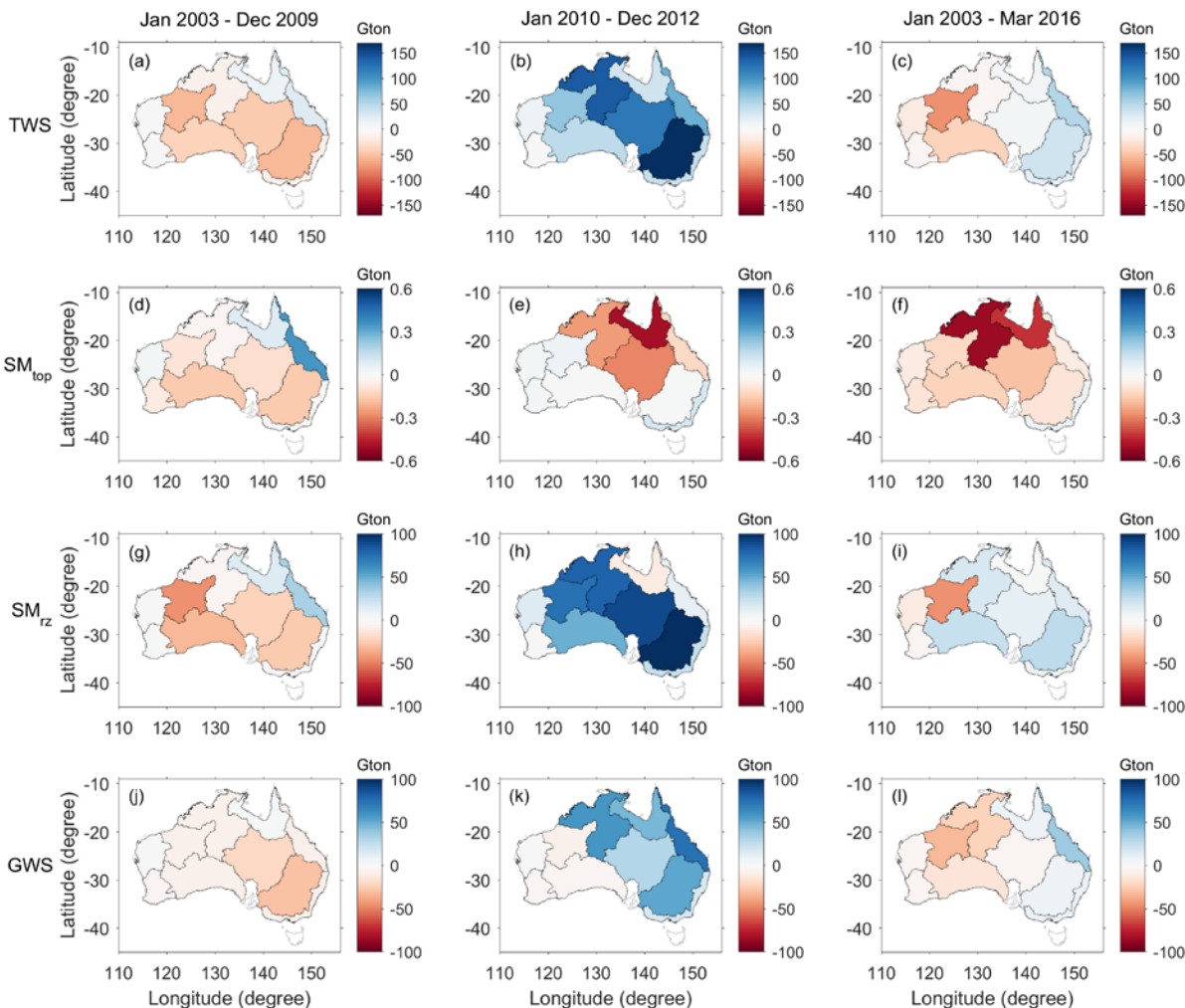


**Figure 10.** Mass changes (Gton, Giga tonne) of $\Delta TWS$, $\Delta SM_{top}$, $\Delta SM_{rz}$, and $\Delta GWS$
estimated from GC solutions over 10 Australian basins in 3 different periods, Big Dry
(January 2003 – December 2009), Big Wet (January 2010 – December 2012), and entire
period (January 2003 – March 2016).

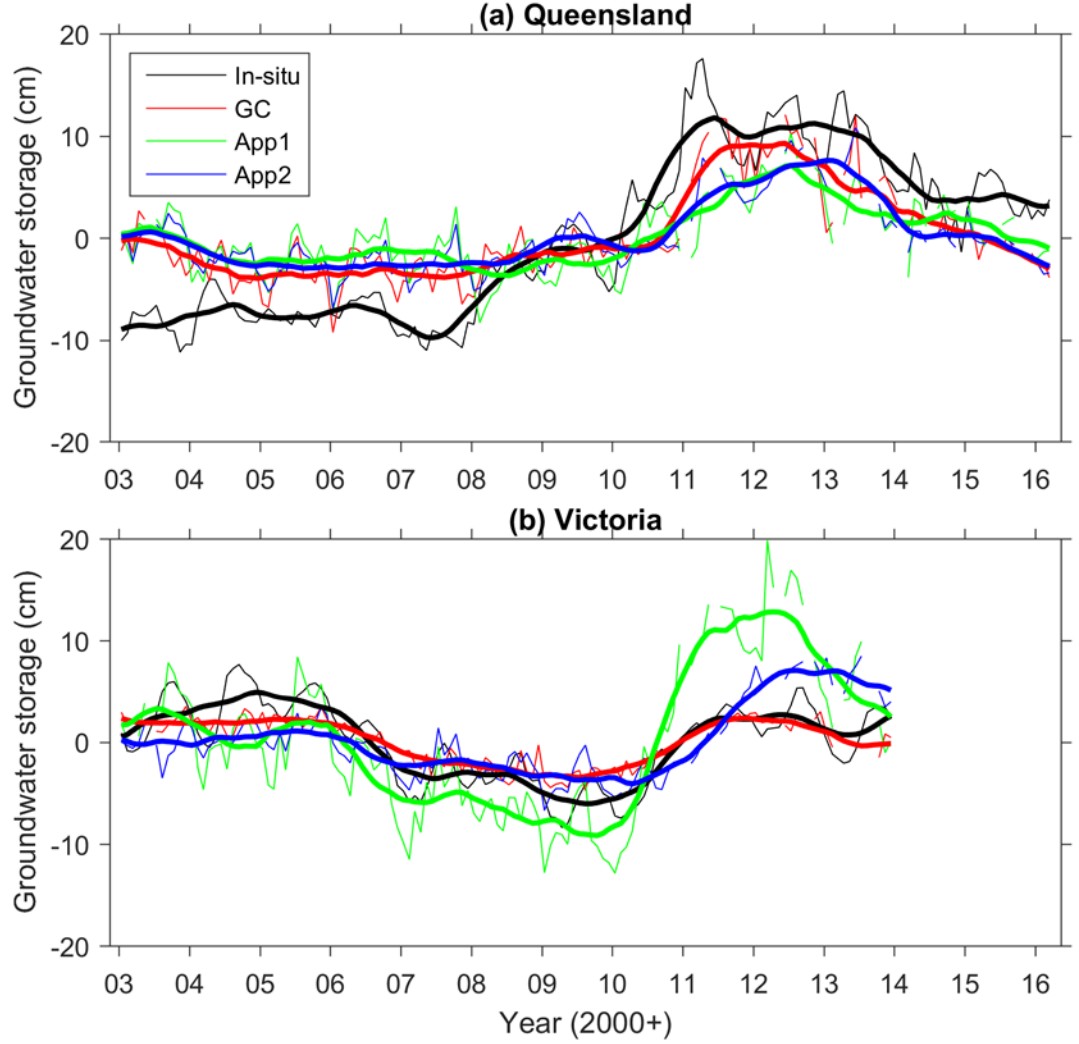


**Figure 11.** $\Delta GWS$ estimated from Approach 1 (App1) and Approach 2 (App2) in Queensland
(a) and Victoria (b). The in-situ groundwater network data and the GC solutions are also
shown. De-seasonalized time series are shown in thick lines.