# Peer review of "On the use of GRACE normal equation of intersatellite tracking data for"

_Hydrology and Earth System Sciences, 2017_

## Referee Comment (RC1) · Anonymous Referee #1 · 4 Jul 2017

The satellite gravimetry mission GRACE is a unique tool to remotely monitor mass transport processes in the Earth System. Temporal changes in gravity are determined from mass anomalies at, above, or beyond the Earth's surface, thus allowing to quantitatively determine water mass losses at global-to-regional scales.

The current paper explores ways to utilize GRACE for the validation of a numerical land surface model over Australia. In contrast to most previous studies, the authors do not utilize high-level processed GRACE data, but start from monthly normal equations as provided by Mayer-Guerr et al. of the University of Graz. Various alternative observational data-sets are utilized to discuss the GRACE results and relate them to the

hydroclimatological conditions of Australia over the last 13 years. The paper appears to be a generally very valuable contribution to the literature and I strongly recommend its further consideration for publication in HESS. A number of points might be, however, considered during a revision in order to improve the presentation.

(1) The description of the GC approach appears to be somewhat ambiguous: While Sect. 2 is claiming to use L1B KBRR data only, it becomes clear from Sect. 3 that in fact L2 monthly normal equations from ITSG2016 are applied. Those NEQ, however, include not only KBRR but all GRACE sensor information (KBRR, ACC, GPS, attitude) and a priori background models (AOD1B, earth, ocean, and atmospheric tides, third body effects). I suggest that comparisons with the official ITSG2016 monthly solutions are included in order to demonstrate the added-value of the GC approach over the standard L2 data. Note that comparisons against GRGS or JPL monthly solutions as already (partly) included in the paper will not be sufficient since ITSG2016 is commonly perceived as a GRACE series of particularly high quality.

(2) The GC approach assumes that model errors are normally distributed with zero mean (eq. 1). Authors should provide more evidence that this assumption is indeed justified in their setting.

(3) line 284: It is optimistic to assume that the model omission error can be fully accounted by just increasing the model covariance by 20%, in particular since this assumes that omission errors do not contribute to biases (which is quite unlikely). More evidence need to be provided for the (approximate) validity of this assumption.

(4) The statement of line 212ff is unclear (and apparently not picked up again in the remainder of the paper). Please elaborate.

(5) Line 289: What does "cooperating" mean in that case?

(6) The specific yield for the Queensland and Victoria networks differ by a factor of 2: Is there any geologic evidence/argument available for those very different yield factors?

(7) Sect. 6.2 appears to be rather an outlook to a future study. Since no actual results are presented, I am wondering if this section should not be better removed entirely?

(8) Major mining activities are currently taking place at the North West Plateau where GRACE picks up negative mass trends: What measures have been taken to reduce mass loss un-related to the terrestrial water cycle from the GRACE data in particular in that area?

---

## Referee Comment (RC2) · Anonymous Referee #2 · 11 Jul 2017

This study combines GRACE's least-squares normal equations of L1B data and results from a hydrological model to improve soil moisture and groundwater estimates. It highlights the importance of the full error variance-covariance information of GRACE data on optimally integrating observations and model estimates. The GRACE-combined approach shows better agreements with in-situ groundwater and soil moisture observations at basin and inter-annual time scales.

This is a well-written article with interesting results. However, I find the methodology is ambiguous. This study claimed the use of raw GRACE L1B data in combination of model outputs. However, the gridded L3 data from GRGS solutions were also used in

the calculation together with the normal equations of GRACE that were obtained from the ITSG-Grace2016 products. Also, the normal equations and gridded TWS were from different centres. Why not using ITSG derived TWS products to be consistent? The validation of soil moisture and groundwater estimates is not sufficient enough to support the conclusion of the article. Overall, this study is valuable for the community. I recommend it for publication after addressing my following concerns.

Line 152: Equation 9 is the most important equation in this study, but some of the information is provided in the later section 3. Also the model covariance matrix is provided in section 4.2. Authors might consider making the method section clearer and reduce some unnecessary equations.

Line 170: Basically, the paper claimed "the use of intersatellite tracking data", but the data was the normal matrix N and vector c obtained from the ITSG-Grace 2016 as well as the gravity field coefficient from GOCO05s solution. No Level 1B data was actually used directly in this study, so I wonder whether the title is appropriate.

Line 195: The GRGS gridded TWS products were used in Equation 9 to work out the TWS values outside Australia. The L3 GRGS products derived from the Earth's geopotential coefficients up to degree and order 80, while ITSG data used in the study were up to 90. Why not using the ITSG TWS data? Can the ITSG normal equation represent the uncertainty in L3 GRGS products?

Line 210: The gridded GRGS data was resampled to 0.5 degree spatially, but the normal equation only contains the information to degree 90. How did you deal with the different spatial scale in the error variance-covariance matrix?

Line 231: depth between 0.022 m not cm

Line 256: The sensitivity study of the model parameters is an important process but not necessary to show in the paper. Author may consider removing table 2.

Line 298: Did you do the CDF matching for few years and validate the results for the rest

of time period? Or did you match all the time series and validate the same time period? If so, your estimates and observations are not independent. The CDF matching may discard important signals of the observations. Since only correlation was calculated, CDF matching is not necessary.

Line 301: The variability of soil moisture inside a basin is quite high. The average of basin and monthly soil moisture can smooth out lot of signals. Since your output is $0.5°$x $0.5°$ gridded products, why not validate at this scale instead of basin scale? Can you show some validation with in-situ measurements?

Line 307: The groundwater estimates were only validated for two states using the state average. It should be possible to validate all the states over Australia or at basin scale to be consistent with other results. Two states are not sufficient to support the improvement in groundwater storage estimates over Australia.

Line 320: Is that only one value of specific yield per state was used to convert the groundwater level to storage? Will it be more appropriate to use different specific yields for different locations and calculate the average?

Line 468: The difference between model and GC approach for soil moisture is marginal here for basin monthly average. Can you show some time series examples of GC results and AMSR-E retrievals?

Line 482: In Table 4, the trend of GC approach is about one third of in-situ measurements for Queensland. What causes such big difference?

Line 491: It will be interesting to see the groundwater storage change in Murray-Darling Basin after the GC approach compared with in-situ measurements, during the big drought and big wet period.

Line 502: This section investigates the mass variation in the past 13 years based on the GC approach. Figure 8 is a quite good demonstration of the mass variation at different layers of water storage. The top and root-zone soil moisture show quite different

trends. The root-zone soil moisture has similar trends with TWS and groundwater for most of the basins. It will be better to have some validation of root-zone soil moisture estimates and more sufficient groundwater storage estimates to support the analysis in this section.

Line 579: 0.39 is App1 and 0.42 is App2? So the trend calculating from GRACE subtracting modeled soil moisture is the same with modeled groundwater trend (in Table 4). The NS value for App1 is 0.46, which is less than the CABLE model without GRACE? For Victoria, the NS value of App2 is 0.3 less than CABLE model without GRACE too. With the assimilation of GRACE in App2, the correlation is degraded. It seems model itself without GRACE is better compared to App1 or App2. Still, for only two states validation results, it's hard to demonstrate that GC approach works better due to the error information. It could be your model uncertainty is better estimated using the ensembles as explained in Section 4.2. When you do the App1 and App2, did you also used 7 precipitation dataset as the same as the GC approach? Please clarify.

Line 587: The future work in this section is interesting but no results were provided. Author may consider removing this section completely or providing the results in this paper together with the GC approach.
* * *

---

## Referee Comment (RC3) · Anonymous Referee #3 · 26 Jul 2017

I read the paper by Tangdamrongsub et al. with an interest. Most of researches to date address the recovery of time-variable gravity fields from GRACE level-1 observations. These data are then used to estimate total water storage (TWS) changes, which represent a vertical summation of mass changes within different compartments including the Earth's surface and its sub-surface. Although, having an access to TWS is a unique opportunity and therefore these data have been used to study regional and global mass redistribution, separated estimation of water storage as surface water storage, soil water storage and groundwater quantities are of interest of many hydrological and water resources studies. This study suggests an inversion approach to directly estimate the contribution of water storage in soil and groundwater, while inverting GRACE level 1

measurements. The idea of this paper is very good and has been somewhat a 'dream' since starting the GRACE mission. However, there are many technical issues that have not been correctly addressed here, which prevent me to recommend a positive decision. In the following my major concerns are listed:

L15-L16–> This is not true that "there is no covariance matrix for L2 products". After filtering and conversion to TWS, this error can be propagated, which is addressed e.g., in DOI:10.1007/s10712-014-9309-8. L48 repeats the same claim!

L16-L17: The consequence is undesired alteration of ... data and its statistical property. –> It is not clear what this means. Are you suggesting that all other published papers are wrong!?

L21-L22–> This is not clear which approach has been used.

L61-L64–> Inversion techniques for signal separation have been applied, which consider errors in GRACE and complementary data used for signal separation. DOI:10.1016/j.jog.2012.03.001; DOI:10.1016/j.jog.2011.02.003; DOI:10.1007/s10712-016-9403-1

The Methodology section needs to be specified, please add appendices to clearly how the equations are built. I cannot figure out how the normal equation is formulated, whether it includes KBRR and any orbital information? The accuracy of recovery has not been justified, which is essential for any scientific appliction to show that the accuracy of software is comparable with official products. Please include comparisons with the official ITSG2016 monthly solutions. L120–> Please describe how the matrix A is derived and what are the entries. Similarly L128-L130 are unclear.

Results of the inversion might be compared with those that assimilate GRACE into hydrological models to improve the surface/sub-surface storage compartments. Recent studies over Australia include: DOI:10.1002/2016WR019641; DOI:10.1016/j.advwatres.2017.07.001

---

## Author Comment (AC1) · 31 Aug 2017

We would like to acknowledge the insightful comments and suggestions provided by reviewer 1. We will consider the reviewer's suggestions in our revised manuscript. Followings are the responses (R) based on the comments:

(1) The description of the GC approach appears to be somewhat ambiguous: While Sect. 2 is claiming to use L1B KBRR data only, it becomes clear from Sect. 3 that in fact L2 monthly normal equations from ITSG2016 are applied. Those NEQ, however, include not only KBRR but all GRACE sensor information (KBRR, ACC, GPS, attitude) and a priori background models (AOD1B, earth, ocean, and atmospheric tides, third

[Figure]

Creative Commons CC BY license logo

body effects).

R1: Reviewer is correct that the normal equation is built including measurements from all GRACE sensors, including KBR, and a priori background models. Therefore, we will rephrase lines 121-122 as follows: "... the observation vector containing various kinds of L1B data including the inter-satellite ranging"

I suggest that comparisons with the official ITSG2016 monthly solutions are included in order to demonstrate the added-value of the GC approach over the standard L2 data. Note that comparisons against GRGS or JPL monthly solutions as already (partly) included in the paper will not be sufficient since ITSG2016 is commonly perceived as a GRACE series of particularly high quality.

R2: We thank for reviewer suggestion. However, the official ITSG2016 solution is the unconstraint gravity field. Deriving TWS from the unconstraint SHC requires filtering, which might lead to the alteration of the GRACE signal. Comparing the GC result with the filtered GRACE data may lead to misinterpretation of the GC performance. Therefore, we do not compare our GC result with the ITSG2016 solution. Instead, we compare the GC result with the independent GRACE-derived TWS product such as the GRGS and Mascon that do not require further post-processing.

(2) The GC approach assumes that model errors are normally distributed with zero mean (eq. 1). Authors should provide more evidence that this assumption is indeed justified in their setting.

R3: The GC approach is developed based on the least-square combination, which assumes the uncertainty following the normal distribution with zero mean and covariance C. The derivation and setting of model uncertainty under the given assumption (e.g., zero mean) is described in Sect. 4.2 of the submitted manuscript.

(3) line 284: It is optimistic to assume that the model omission error can be fully accounted by just increasing the model covariance by 20%, in particular since this assumes that omission errors do not contribute to biases (which is quite unlikely). More evidence need to be provided for the (approximate) validity of this assumption.

R4: It is difficult to acquire a precise omission error from CABLE. This might require an extensive experiment, which is a subject of independent study. Therefore, we simply assumed the omission error base on trial-and-error to be a good compromise between increasing of the model error (due to the omission error) and not exceeding the TWS error suggested by Dumedah and Walker (2014). Note that, this is one way to construct the error statistic and we understand the limitation of such an assumption. Therefore, we will consider including the statement regarding the limitation of this approach in the revised manuscript.

(4) The statement of line 212ff is unclear (and apparently not picked up again in the remainder of the paper). Please elaborate.

R5: In this section, we describe the reason and the usage of the independent GRACE solutions. The GC approach requires the knowledge of the \DeltaTWS outside the study area, and the GRGS is used for this purpose. This is clearly stated in lines 213 – 216: "To obtain the \DeltaTWS values outside Australia. As shown in Eq. (9), .. the GRGS solutions as the GRGS solution provides \DeltaTWS at a spatial resolution comparable to the normal equation data." The GRGS and mascon solutions are also used for the comparison purpose (lines 217 – 218), and the application of both solutions have been discussed in the paper e.g., Sect. 5.2.1, Sect. 6.1.

(5) Line 289: What does "cooperating" mean in that case?

R6: Replaced with "using" in the revised manuscript.

(6) The specific yield for the Queensland and Victoria networks differ by a factor of 2: Is there any geologic evidence/argument available for those very different yield factors?

R7: We obtained the specific yield values from the published literatures (Chen et al., 2016; Rassam et al., 2013, Welsh 2008). Unfortunately, there is no record about the

geological structure reported in those references.

(7) Sect. 6.2 appears to be rather an outlook to a future study. Since no actual results are presented, I am wondering if this section should not be better removed entirely?

R8: We agree with reviewer. Section 6.2 will be removed from the manuscript.

(8) Major mining activities are currently taking place at the North West Plateau where GRACE picks up negative mass trends: What measures have been taken to reduce mass loss un-related to the terrestrial water cycle from the GRACE data in particular in that area?

R9: This has been discussed in the past few years. It is more likely that the negative mass trends is mainly caused by the declining rainfall after 2000 (van Dijk et al., 2011), and unlikely mining (please see also http://www.news.com.au/technology/environment/climate-change/nasa-study-says-the-canning-basin-in-wa-is-being-depleted-too-fast/news-story/9bf107b8299c19b57904ed719639a0ba). The hydrological signal tends to be much larger than the mining signal. This is supported by model's water storage estimate that describes more than 90-95% of GRACE signal (see e.g., TRMM in Fig. 2). Such a small signal (from mining) is unlikely picked up by GRACE. Therefore, we do not take the mining signal into our calculation.

Reference

van Dijk, A. I. J. M., L. J. Renzullo, and M. Rodell (2011), Use of Gravity Recovery and Climate Experiment terrestrial water storage retrievals to evaluate model estimates by the Australian water resources assessment system, Water Resour. Res., 47, W11524, doi:10.1029/2011WR010714.

---

## Author Comment (AC2) · 31 Aug 2017

We would like to acknowledge the insightful comments and suggestions provided by reviewer 2. We will consider the reviewer's suggestions in our revised manuscript. Followings are the responses (R) based on the comments:

Line 152: Equation 9 is the most important equation in this study, but some of the information is provided in the later section 3. Also the model covariance matrix is provided in section 4.2. Authors might consider making the method section clearer and reduce some unnecessary equations.

R1: We thank for the reviewer's suggestions. We organize the paper's content in this way to separate the methodology and implementation of the proposed approach. Section 2 describes the methodology of the GRACE-GC in general, which can be simply applied to any GRACE data or land surface model. Later on, sections 3 and 4 describes the specific implementation with the ITSG data and CABLE model, respectively. We believe this is a logical presentation of our methods. This is explained in the manuscript lines 94 – 98: "Firstly, the derivation of GC approach is presented in Sect. 2 while the description of GRACE data processing, including the use of GRACE normal equation is given in Sect. 3. Secondly, the CABLE modelling is outlined in Sect. 4. This includes the derivation of model uncertainty based on the quality of precipitation data and the model parameter inputs."

Line 170: Basically, the paper claimed "the use of intersatellite tracking data", but the data was the normal matrix N and vector c obtained from the ITSG-Grace 2016 as well as the gravity field coefficient from GOCO05s solution. No Level 1B data was actually used directly in this study, so I wonder whether the title is appropriate.

R2: We agree with reviewer. Although the normal equation is constructed based on the L1B data (see lines 172 – 174 in the submitted manuscript, "All L1B data including KBR inter-satellite tracking data, attitude, accelerometer, GPS based kinematic orbit data and AOD1B corrections are reduced in terms of the normal equations"), we understand that the title might be misleading. Therefore, based on reviewer's suggestion, we will consider changing the paper title to "On the use of GRACE normal equation for improved estimation of soil moisture and groundwater in Australia" in the revised manuscript.

Line 195: The GRGS gridded TWS products were used in Equation 9 to work out the TWS values outside Australia. The L3 GRGS products derived from the Earth's geopotential coefficients up to degree and order 80, while ITSG data used in the study were up to 90. Why not using the ITSG TWS data? Can the ITSG normal equation represent the uncertainty in L3 GRGS products?

R3: Eq. (9) requires the knowledge of \DeltaTWS outside the study region. However, the ITSG does not provide such a solution (L3). The official ITSG2016 solution is the unconstraint gravity field. Deriving \DeltaTWS from the unconstraint SHC requires filtering, which might lead to the alteration of the GRACE signal. This is the main reason we use the GRGS solution for such a purpose.

Line 210: The gridded GRGS data was resampled to 0.5 degree spatially, but the normal equation only contains the information to degree 90. How did you deal with the different spatial scale in the error variance-covariance matrix?

R4: This study uses the least-square combination approach in spatial domain. It is possible to compute the \DeltaTWS from the normal equation at any spatial scale (here is 0.5 degree). However, it is noted that these 0.5 degree data are spatial correlated, and the correlated error is already accounted in the GC approach described in Sect. 2.

Line 231: depth between 0.022 m not cm

R5: Reviewer is correct. We will correct 0.022 cm to 0.022 m in the revised manuscript.

Line 256: The sensitivity study of the model parameters is an important process but not necessary to show in the paper. Author may consider removing table 2.

R6: We thank for reviewer's suggestion. However, we find Table 2 contains some insight information about the model, which are useful for the readers. Therefore, we will keep Table 2 in the revised manuscript.

Line 298: Did you do the CDF matching for few years and validate the results for the rest of time period? Or did you match all the time series and validate the same time period? If so, your estimates and observations are not independent. The CDF matching may discard important signals of the observations. Since only correlation was calculated, CDF matching is not necessary.

R7: We built the CDF using 2003-2004 data and applied it to the rest. As the NS coefficient (not correlation coefficient) is used in this section, the remaining bias might

result in poor NS values, the bias correction is then necessary. For clarity, we will add the additional statement in to the revised manuscript as follows: "The CDF is built using the 2003-2004 data, and it is used for the entire time period"

Line 301: The variability of soil moisture inside a basin is quite high. The average of basin and monthly soil moisture can smooth out lot of signals. Since your output is 0.5_x 0.5_ gridded products, why not validate at this scale instead of basin scale? Can you show some validation with in-situ measurements?

R8: We thank for reviewer's suggestion. We also conducted the analysis at 0.5x0.5 scale, and it provides very similar conclusions to the basin mean. Both results do not show the significant improvement, which is supported by the recent publication (Tian, et al., 2017). Moreover, we did not have an access to the in situ data by the time of this study. Therefore, the validation with in-situ measurements is not conducted in this paper.

Line 307: The groundwater estimates were only validated for two states using the state average. It should be possible to validate all the states over Australia or at basin scale to be consistent with other results. Two states are not sufficient to support the improvement in groundwater storage estimates over Australia.

R9: We agree with reviewer that the validation over all the states is important. However, the ground observation network of Australia is very sparse, and particularly we did not have an access to the data of other states by the time of this study. Therefore, we only validate the result in all possible states of Australia, Queensland and Victoria.

Line 320: Is that only one value of specific yield per state was used to convert the groundwater level to storage? Will it be more appropriate to use different specific yields for different locations and calculate the average?

R10: We agree with reviewer that it is ideal to conduct the conversion at all grid cell independently. However, such information is unfortunately not available. Therefore, we
exploit the best knowledge by obtaining the values from the published literatures for the conversion.

Line 468: The difference between model and GC approach for soil moisture is marginal here for basin monthly average. Can you show some time series examples of GC results and AMSR-E retrievals?

R11: Below are the examples of the time series between our estimates and AMSR-E (C-band): [Figure 1] The statistical value can be found in Table 3 of the submitted manuscript. As stated in the manuscript, no significant change is seen in GC solution, likely due to limitation of GRACE's temporal and spatial resolution. This same conclusion was reported by Tian et al. (2017). Therefore, we decide not to discuss it further in our manuscript. Instead, we provide a reference and recommendation how SM estimates can be improved in the conclusion section of the submitted manuscript.

Line 482: In Table 4, the trend of GC approach is about one third of in-situ measurements for Queensland. What causes such big difference?

R12: As a matter of fact that the GC approach optimally combines the GRACE observation with the model results, the GC result is inevitably influenced by the model estimate. As seen in Fig. 7 and 9, the GC result moves toward the in situ measurement but it is still influenced by the \DeltaGWS estimation from the model.

Line 491: It will be interesting to see the groundwater storage change in Murray-Darling Basin after the GC approach compared with in-situ measurements, during the big drought and big wet period.

R13: The reviewer's suggestion is very interesting. However, we did not have the in situ groundwater data of the Murray-Darling Basin by the time of this study. We will consider the validation in the Murray-Darling Basin in our future study.

Line 502: This section investigates the mass variation in the past 13 years based on the GC approach. Figure 8 is a quite good demonstration of the mass variation at different layers of water storage. The top and root-zone soil moisture show quite different trends. The root-zone soil moisture has similar trends with TWS and groundwater for most of the basins. It will be better to have some validation of root-zone soil moisture estimates and more sufficient groundwater storage estimates to support the analysis in this section.

R14: We appreciate reviewer suggestion and we strongly agree. However, similar to what we stated above, we did not have an access to the in situ root-zone soil moisture of Australia, and the validation of such a component was not possible.

Line 579: 0.39 is App1 and 0.42 is App2? So the trend calculating from GRACE subtracting modeled soil moisture is the same with modeled groundwater trend (in Table 4). The NS value for App1 is 0.46, which is less than the CABLE model without GRACE? For Victoria, the NS value of App2 is 0.3 less than CABLE model without GRACE too. With the assimilation of GRACE in App2, the correlation is degraded. It seems model itself without GRACE is better compared to App1 or App2. Still, for only two states validation results, it's hard to demonstrate that GC approach works better due to the error information. It could be your model uncertainty is better estimated using the ensembles as explained in Section 4.2. When you do the App1 and App2, did you also used 7 precipitation dataset as the same as the GC approach? Please clarify.

R15: Reviewer is correct that App1 and App2 provide poorer results compared to CA-BLE (in some case) or GC approach. However, this section demonstrates the scenario when different \DeltaGWS computation approach are used (not GC). The model part remains the same (as in GC approach). App1 uses mascon solution with error free scenario while App2 uses mascon with its variance matrix, and the different outputs are mainly attributed to the different application of the uncertainty type. The poor results of App1 and App2 are mainly due to too simplified error information implemented. For clarity, we will add the additional statement in the revised manuscript as follows: "The model uncertainty remains the same as in GC approach (Sect. 4.2). The different

outputs between App1 and App2 are mainly attributed to the different application of the uncertainty type."

Line 587: The future work in this section is interesting but no results were provided. Author may consider removing this section completely or providing the results in this paper together with the GC approach.

R16: We agree with reviewer. Section 6.2 will be removed from the manuscript.

[Figure]

Fig. 1.

---

## Author Comment (AC3) · 31 Aug 2017

We would like to acknowledge the insightful comments and suggestions provided by reviewer 3. We will consider the reviewer's suggestions in our revised manuscript. Followings are the responses (R) based on the comments:

L15-L16–> This is not true that "there is no covariance matrix for L2 products". After filtering and conversion to TWS, this error can be propagated, which is addressed e.g., in DOI:10.1007/s10712-014-9309-8. L48 repeats the same claim!

R1: Review is correct that the covariance matrix is available for L2 product. However,

the gridded product (Level 3, not L2 product) is discussed here. This is clearly written in L13-L16 of the submitted manuscript: "... from the high level products (e.g., land grid). The gridded data products are subjected to several drawbacks such as signal attenuation and/or distortion caused by ad hoc posteriori filters, and a lack of error covariance information." For clarity, we will include the additional information in the revised manuscript as follows: "... from the high level products (e.g., land grid from the Level 3 product). The gridded data products ..."

L16-L17: The consequence is undesired alteration of ... data and its statistical property. It is not clear what this means. Are you suggesting that all other published papers are wrong!?

R2: We mean that the post-processing process of GRACE data might lead to the undesired alteration of the signal and its statistical property. To avoid the confusion, we will modify the statement in the revised manuscript as follows: "The post-processing process of GRACE data might lead to the undesired alteration of the signal and its statistical property"

L21-L22–> This is not clear which approach has been used.

R3: The approach used in this study is the least-squares combination. This is clearly mentioned in the earlier sentence (L19-L22): "The approach combines the GRACE's least-squares normal equation (full error variance-covariance information) of L1B data with the results from the Community Atmosphere Land Exchange (CABLE) model to improve soil moisture and groundwater estimates."

L61-L64–> Inversion techniques for signal separation have been applied, which consider errors in GRACE and complementary data used for signal separation. DOI:10.1016/j.jog.2012.03.001; DOI:10.1016/j.jog.2011.02.003; DOI:10.1007/s10712-016-9403-1

R4: We thank reviewer for the suggested literature. We will include them in the revised

manuscript as follows: "Several signal separation techniques have been developed, which considered the errors in GRACE and complementary data in the signal separation process (Rietbroek et al., 2012; Schmeer et al., 2012; Forootan et al., 2017). However, the GRACE uncertainty is commonly derived empirically not necessarily reflecting the true GRACE error characteristics. Similar issue is seen in the data assimilation application (e.g., Zaitchik et al., 2008; Tangdamrongsub et al., 2015; Tian et al., 2017)."

The Methodology section needs to be specified, please add appendices to clearly how the equations are built. I cannot figure out how the normal equation is formulated, whether it includes KBRR and any orbital information? L120–> Please describe how the matrix A is derived and what are the entries. Similarly L128-L130 are unclear.

R5: In this study, the normal equation from the ITSG-2016 is used, and the description of the data can be found in the data webpage (https://www.tugraz.at/institute/ifg/downloads/gravity-field-models/itsg-grace2016). As the derivation of the normal equation is not the focus of this study, we do not discuss it further but refer to the description in the data webpage and references therein (this is stated clearly in Sect. 3.1). The element of the matrix A is mainly the partial derivative of the variational equation respect to the orbital information and gravitational coefficients. The variational equation include both orbital information and various kinds of L1B data including KBR data.

The accuracy of recovery has not been justified, which is essential for any scientific appliction to show that the accuracy of software is comparable with official products. Please include comparisons with the official ITSG2016 monthly solutions.

R6: The objective of this study is to combine GRACE normal equation with land surface model result to improve the estimated SM and GWS, and do not independently resolve the GRACE solution. This is stated clearly in the introduction lines 80 – 82 in the submitted manuscript: "The approach optimally combines the GRACE's least-squares

normal equations with CABLE to improve \DeltaSM and \DeltaGWS estimates."

Therefore, the accuracy of the estimated result is only compared with the model-only result (please see Fig. 6). The accuracy of our TWS estimate can reach < 2 cm, which is in line with the GRACE accuracy of $\sim$ 2 cm globally (Wahr et al., 2006).

Furthermore, the official ITSG2016 solution is the unconstraint gravity field. Deriving TWS from the unconstraint SHC requires filtering, which might lead to the alteration of the GRACE signal. Comparing the GC result with the filtered GRACE data might lead to misinterpretation of the GC performance. Therefore, we do not compare our GC result with the ITSG2016 solution. Instead, we compare the GC result with the independent GRACE-derived TWS product such as the GRGS and Mascon that do not require further post-processing.

Results of the inversion might be compared with those that assimilate GRACE into hydrological models to improve the surface/sub-surface storage compartments. Recent studies over Australia include: DOI:10.1002/2016WR019641; DOI:10.1016/j.advwatres.2017.07.001

R7: We thank for the reviewer suggestion. However, the suggested literature were not published when this manuscript is finalized/under reviewed. We will consider the reviewer suggestion in our future work.

References

Forootan, E., Safari, A., Mostafaie, A., Schumacher, M., Delavar, M., and Awange, J. L.: Large-Scale Total Water Storage and Water Flux Changes over the Arid and Semi-arid Parts of the Middle East from GRACE and Reanalysis Products, Surv. Geophys., 38:591-615, doi:10.1007/s10712-016-9403-1, 2017.

Rietbroek, R., Fritsche, M., Brunnabend, S.-E., Daras, I., Kusche, J., Schröter, J., Flechtner, F., and Dietrich, R.: Global surface mass from a new combination of GRACE, modelled OBP and reprocessed GPS data, J. Geodyn., 59 - 60:64 - 71,

doi:10.1016/j.jog.2011.02.003, 2012.

Schmeer, M., Schmidt, M., Boschb, W., and Seitz, F.: Separation of mass signals within GRACE monthly gravity field models by means of empirical orthogonal functions, J. Geodyn., 59 - 60:124 - 132, doi:10.1016/j.jog.2012.03.001, 2012.

---

## Author Response (AR1)

**Reviewer 1**

We would like to acknowledge the insightful comments and suggestions provided by reviewer 1. We will consider the reviewer's suggestions in our revised manuscript. Followings are the responses (R) based on the comments:

(1) The description of the GC approach appears to be somewhat ambiguous: While Sect. 2 is claiming to use L1B KBRR data only, it becomes clear from Sect. 3 that in fact L2 monthly normal equations from ITSG2016 are applied. Those NEQ, however, include not only KBRR but all GRACE sensor information (KBRR, ACC, GPS, attitude) and a priori background models (AOD1B, earth, ocean, and atmospheric tides, third body effects).

**R1:** Reviewer is correct that the normal equation is built including measurements from all GRACE sensors, including KBR, and a priori background models. Therefore, lines **144 – 145** of the revised manuscript is revised as follows:

"… the observation vector containing various kinds of L1B data including the inter-satellite ranging"

I suggest that comparisons with the official ITSG2016 monthly solutions are included in order to demonstrate the added-value of the GC approach over the standard L2 data. Note that comparisons against GRGS or JPL monthly solutions as already (partly) included in the paper will not be sufficient since ITSG2016 is commonly perceived as a GRACE series of particularly high quality.

**R2:** The computation of $\Delta TWS$ from ITSG L2 solutions (like other L2 solutions) is subject to the post-processing filters and often followed by the application of empirical scaling factors, obtained from the land surface models. Our study provides a more rigorous way of computing $\Delta TWS$ without going through such ad hoc procedures. GRGS and JPL mascon are internally or post-processed and it is not expected users to apply (subjective) post-processing. This is the primary reason we validate our results with GRGS and JPL mascon.

(2) The GC approach assumes that model errors are normally distributed with zero mean (eq. 1). Authors should provide more evidence that this assumption is indeed justified in their setting.

**R3:** As the reviewer concerned, the GC approach is developed based on the least-square combination, which assumes the uncertainty following the normal distribution with zero mean and covariance C. The derivation and setting of model uncertainty under the given assumption (e.g., zero mean) and its limitation are clarified in Sect. 4.2. For clarity, we modify lines **298 – 301** of the revised manuscript as follows:

"The GC approach assumes that model errors are normally distributed with zero mean. Any violation of this assumption will yield a bias in the combined solutions. Therefore, the mean value is removed from each ensemble member, $\mathcal{H}_R{}' = \mathcal{H}_R - \tilde{h}_R$, and the error covariance matrix of the model is empirically computed as"

(3) line 284: It is optimistic to assume that the model omission error can be fully accounted by just increasing the model covariance by 20%, in particular since this assumes that omission errors do not contribute to biases (which is quite unlikely). More evidence need to be provided for the (approximate) validity of this assumption.

**R4:** Rigorous development of the statistical property of the land surface models (expectation and covariance) is crucial but a difficult task. The empirical way was taken in this study. As also noted by the reviewer, the omission will likely introduce a bias in the solutions as well.

We acknowledge this in the last paragraph of Section 4.2. Lines **313 – 319** of the revised manuscript now reads:

"It is emphasized that the model omission error caused by imperfect modelling of hydrological process within the LSM is not taken into account in the above description. The omission error may increase the model covariance and introduce a bias as well. We account for the omission error by increasing 20% of the model covariance. (i.e., multiplying $C_R$ by 1.2). We determine such omission error based on trial-and-error such that it increases the model error (due to the omission error) but not exceeds the model error value reported by Dumedah and Walker (2014). We acknowledge that this is only a simple practical way of accounting for the omission error into the total model error."

(4) The statement of line 212ff is unclear (and apparently not picked up again in the remainder of the paper). Please elaborate.

**R5:** This is to introduce what other independent GRACE solutions (GRGS and JPL mascon) to be used to compare our GC results. The solutions are used in Sect. 5.2 and 5.5 in the revised manuscript.

(5) Line 289: What does "cooperating" mean in that case?

**R6:** Replaced with "using" in the revised manuscript.

(6) The specific yield for the Queensland and Victoria networks differ by a factor of 2: Is there any geologic evidence/argument available for those very different yield factors?

**R7:** We took the specific yield values simply from the literatures (Chen et al., 2016; Rassam et al., 2013, Welsh 2008).

(7) Sect. 6.2 appears to be rather an outlook to a future study. Since no actual results are presented, I am wondering if this section should not be better removed entirely?

**R8:** We thank for reviewer's suggestion. Section 6.2 is removed from the manuscript.as suggested.

(8) Major mining activities are currently taking place at the North West Plateau where GRACE picks up negative mass trends: What measures have been taken to reduce mass loss un-related to the terrestrial water cycle from the GRACE data in particular in that area?

**R9:** Reviewer made a good point. However, it is likely that the negative mass trends is mainly caused by the declining rainfall after 2000 (van Dijk et al., 2011), and less likely mining activity, as confirmed by the government WA Department of Water.

http://www.news.com.au/technology/environment/climate-change/nasa-study-says-the-canning-basin-in-wa-is-being-depleted-too-fast/news-story/9bf107b8299c19b57904ed719639a0ba

**Reference**

van Dijk, A. I. J. M., L. J. Renzullo, and M. Rodell (2011), Use of Gravity Recovery and Climate Experiment terrestrial water storage retrievals to evaluate model estimates by the Australian water resources assessment system, Water Resour. Res., 47, W11524, doi:10.1029/2011WR010714.

**Reviewer 2**

We would like to acknowledge the insightful comments and suggestions provided by reviewer 2. We will consider the reviewer's suggestions in our revised manuscript. Followings are the responses (R) based on the comments:

Line 152: Equation 9 is the most important equation in this study, but some of the information is provided in the later section 3. Also the model covariance matrix is provided in section 4.2. Authors might consider making the method section clearer and reduce some unnecessary equations.

**R1:** We organize the paper to introduce mathematical developments and followed by implementation. Section 2 describes the general methodology, while sections 3 and 4 describe the specifics needed to implement the algorithm with the ITSG data and CABLE model, respectively. We believe this is a logical presentation of our methods. This is clarified in the revised manuscript, lines **116 – 120** as follows:

"Firstly, the derivation of GC approach is presented in Sect. 2 while the description of GRACE data processing, including the use of GRACE normal equation is given in Sect. 3. Secondly, the CABLE modelling is outlined in Sect. 4. This includes the derivation of model uncertainty based on the quality of precipitation data and the model parameter inputs."

Line 170: Basically, the paper claimed "the use of intersatellite tracking data", but the data was the normal matrix N and vector c obtained from the ITSG-Grace 2016 as well as the gravity field coefficient from GOCO05s solution. No Level 1B data was actually used directly in this study, so I wonder whether the title is appropriate.

**R2:** As reviewer pointed out correctly, we started from the L1B normal equation data, not directly from L1B data. We revised the titles as follows: "On the use of GRACE normal equation of intersatellite tracking data for improved estimation of soil moisture and groundwater in Australia".

Line 195: The GRGS gridded TWS products were used in Equation 9 to work out the TWS values outside Australia. The L3 GRGS products derived from the Earth's geopotential coefficients up to degree and order 80, while ITSG data used in the study were up to 90. Why not using the ITSG TWS data? Can the ITSG normal equation represent the uncertainty in L3 GRGS products?

**R3:** Eq. (9) requires the knowledge of ΔTWS grid outside the study region. However, the ITSG does not provide such gridded data directly. GRGS solution is one of the best possible candidate suitable for correcting the outside effect since it is most compatible with normal equation dataset we use in terms of a resolution and it does not require the (subjective) post-processing of the solutions unlike ITSG and other L2 solutions. This is clarified in lines **240 – 241** of the revised manuscript.

Line 210: The gridded GRGS data was resampled to 0.5 degree spatially, but the normal equation only contains the information to degree 90. How did you deal with the different spatial scale in the error variance-covariance matrix?

**R4:** This study uses the least-square combination approach in spatial domain. It is possible to compute the ΔTWS from the normal equation at any spatial scale (here is 0.5 degree). However, it is noted that the 0.5 degree data are spatial correlated (because of 90 degree resolution), and such correlation in the 0.5 degree grid is already accounted in the GC approach described in Sect. 2.

Line 231: depth between 0.022 m not cm

**R5:** Reviewer is correct. We correct 0.022 cm to 0.022 m in the revised manuscript.

Line 256: The sensitivity study of the model parameters is an important process but not necessary to show in the paper. Author may consider removing table 2.

**R6:** We thank for reviewer's suggestion. We still believe that Table 2 is worth to present since it gives an insight about the model parameters in a direct relevance to the storage outputs. Therefore, we decide to keep Table 2 in the revised manuscript.

Line 298: Did you do the CDF matching for few years and validate the results for the rest of time period? Or did you match all the time series and validate the same time period? If so, your estimates and observations are not independent. The CDF matching may discard important signals of the observations. Since only correlation was calculated, CDF matching is not necessary.

**R7:** We built the CDF using 2003-2004 data and applied it to the rest. As the NS coefficient (not correlation coefficient) is used in this section, the remaining bias might result in poor NS values, the bias correction is then necessary. For clarity, we added the additional statement in lines **335 – 336** of the revised manuscript as follows:

"The CDF is built using the 2003-2004 data, and it is used for the entire period"

Line 301: The variability of soil moisture inside a basin is quite high. The average of basin and monthly soil moisture can smooth out lot of signals. Since your output is 0.5_x 0.5_ gridded products, why not validate at this scale instead of basin scale? Can you show some validation with in-situ measurements?

**R8:** We indeed conducted the analysis at 0.5x0.5 scale (for surface SM), and it provides very similar conclusions to the basin mean. Both results do not show the significant improvement for the surface soil moisture computation, which is supported by the recent publication (Tian, et al., 2017). We discussed why this is the case and the limitation of GRACE data in the original manuscript.

Line 307: The groundwater estimates were only validated for two states using the state average. It should be possible to validate all the states over Australia or at basin scale to be consistent with other results. Two states are not sufficient to support the improvement in groundwater storage estimates over Australia.

**R9:** We agree with reviewer. However, the primary purpose is to demonstrate a statistically rigorous way of using GRACE datasets (not high level processed data) for combining with the LSM models and eventually for data assimilation by emphasizing the importance of using error propagation and full covariance. The ground observation network of Australia is relatively sparse particularly at the state-wide level. Our study is initially based on what we could get.

Line 320: Is that only one value of specific yield per state was used to convert the groundwater level to storage? Will it be more appropriate to use different specific yields for different locations and calculate the average?

**R10:** We agree with reviewer that it is ideal to conduct the conversion at all grid cell independently. However, no such information is available at the grid cell level. We used the best possible knowledge of specific yield from the published literatures.

Line 468: The difference between model and GC approach for soil moisture is marginal here for basin monthly average. Can you show some time series examples of GC results and AMSR-E retrievals?

**R11:** Below are the examples of the time series between our estimates and AMSR-E (C-band):

[Figure]

The statistical value can be found in **Table 3** of the revised manuscript. As stated in the manuscript, no significant change is seen in GC solution, likely due to limitation of GRACE's temporal and spatial resolution. This same conclusion was reported by Tian et al. (2017). We provide a reference and recommendation how SM estimates can be improved in the conclusion section, lines **704 – 710** of the revised manuscript.

"it is challenging to improve surface soil moisture varying rapidly in time, using a monthly mean GRACE observation. Tian et al. (2017) utilized the satellite soil moisture observation from the Soil Moisture and Ocean Salinity (SMOS, Kerr et al., 2001) in addition to GRACE data for their data assimilation and showed a clear improvement in the top soil moisture estimate. The GC approach with complementary observations at higher temporal resolution should be considered particularly to enhance the surface soil moisture computation."

Line 482: In Table 4, the trend of GC approach is about one third of in-situ measurements for Queensland. What causes such big difference?

**R12:** As a matter of fact that the GC approach optimally combines the GRACE observation with the model results, the GC result is inevitably influenced by the model estimate. As seen in Fig. 7 and 9, the GC result moves toward the in situ measurement but it is still influenced by the ΔGWS estimation from the model.

Line 491: It will be interesting to see the groundwater storage change in Murray-Darling Basin after the GC approach compared with in-situ measurements, during the big drought and big wet period.

**R13:** The reviewer's suggestion is very interesting. However, we did not have the in situ groundwater data of the Murray-Darling Basin by the time of this study and so we conducted the analysis based on what we could obtain. We will consider the validation in the Murray-Darling Basin in our future study.

Line 502: This section investigates the mass variation in the past 13 years based on the GC approach. Figure 8 is a quite good demonstration of the mass variation at different layers of water storage. The top and root-zone soil moisture show quite different trends. The root-zone soil moisture has similar trends with TWS and groundwater for most of the basins. It will be better to have some validation of root-zone soil moisture estimates and more sufficient groundwater storage estimates to support the analysis in this section.

**R14:** We appreciate reviewer suggestion and we strongly agree. However, similar to what we stated above, we did not have an access to the in situ root-zone soil moisture of Australia, and the validation of such a component was not possible. We would like to reiterate that this is the first demonstration of new technique of using more fundamental dataset from GRACE, applied to the Australian continent.

Line 579: 0.39 is App1 and 0.42 is App2? So the trend calculating from GRACE subtracting modeled soil moisture is the same with modeled groundwater trend (in Table 4). The NS value for App1 is 0.46, which is less than the CABLE model without GRACE? For Victoria, the NS value of App2 is 0.3 less than CABLE model without GRACE too. With the assimilation of GRACE in App2, the correlation is degraded. It seems model itself without GRACE is better compared to App1 or App2. Still, for only two states validation results, it's hard to demonstrate that GC approach works better due to the error information. It could be your model uncertainty is better estimated using the ensembles as explained in Section 4.2. When you do the App1 and App2, did you also used 7 precipitation dataset as the same as the GC approach? Please clarify.

**R15:** Reviewer is correct that App1 and App2 provide poorer results compared to CABLE (in some case) or GC approach. However, this section demonstrates the scenario when different ΔGWS computation approach are used (not GC). The model part remains the same (as in GC approach). App1 uses mascon solution with error free scenario while App2 uses mascon with its variance matrix, and the different outputs are mainly attributed to the different application of the uncertainty type. The poor results of App1 and App2 are mainly due to too simplified error information implemented. This has been clarified in the revised manuscript lines **630 – 632**:

"Note that the model uncertainty remains the same as in GC approach (Sect. 4.2). The different outputs between App1 and App2 are mainly attributed to the different application of the different estimates of the uncertainty."

Line 587: The future work in this section is interesting but no results were provided. Author may consider removing this section completely or providing the results in this paper together with the GC approach.

**R16:** We thank for reviewer's suggestion. Section 6.2 is removed from the manuscript.as suggested.

**Reviewer 3**

We would like to thank reviewer 3. Followings are the responses (R) based on the comments:

L15-L16–> This is not true that "there is no covariance matrix for L2 products". After filtering and conversion to TWS, this error can be propagated, which is addressed e.g., in DOI:10.1007/s10712-014-9309-8. L48 repeats the same claim!

**R1:** Review is correct that the covariance matrix is available for L2 product. However, it is not the case for the gridded product (Level 3) discussed here. This is clearly written in lines **13 – 16** of the submitted manuscript:

"… from the high level products (e.g., land grid). The gridded data products are subjected to several drawbacks such as signal attenuation and/or distortion caused by ad hoc posteriori filters, and a lack of error covariance information."

For clarity, we include the additional information in line **16** of the revised manuscript as follows:

"… from the high level products (e.g., land grid from the Level 3 product). The gridded data products …"

L16-L17: The consequence is undesired alteration of ... data and its statistical property. It is not clear what this means. Are you suggesting that all other published papers are wrong!?

**R2:** The post-processed (high level grid) data are often used without a proper statistical information. It changes information GRACE data provides. For clarity, we revised the statement as follows:

"The post-processing process of GRACE data might lead to the undesired alteration of the signal and its statistical property."

L21-L22–> This is not clear which approach has been used.

**R3:** The approach used in this study is the least-squares combination. This is clearly mentioned in the earlier sentence (lines **19 – 22** of the submitted manuscript):

"The approach combines the GRACE's least-squares normal equation (full error variance-covariance information) of L1B data with the results from the Community Atmosphere Land Exchange (CABLE) model to improve soil moisture and groundwater estimates."

L61-L64–> Inversion techniques for signal separation have been applied, which consider errors in GRACE and complementary data used for signal separation. DOI:10.1016/j.jog.2012.03.001; DOI:10.1016/j.jog.2011.02.003; DOI:10.1007/s10712-016-9403-1

**R4:** Based on the suggested literatures, the text was revised as follows.

"Several techniques have been developed to separate different signals considering the errors in GRACE and other data (Rietbroek et al., 2012; Schmeer et al., 2012; Forootan et al., 2017). However, the GRACE uncertainty is often derived empirically, not necessarily reflecting the actual GRACE error characteristics. The empirical GRACE errors have been also used in the data assimilation (e.g., Zaitchik et al., 2008; Tangdamrongsub et al., 2015; Tian et al., 2017)."

The Methodology section needs to be specified, please add appendices to clearly how the equations are built. I cannot figure out how the normal equation is formulated, whether it includes KBRR and any orbital information? L120–> Please describe how the matrix A is derived and what are the entries. Similarly L128-L130 are unclear.

**R5:** As written in the original manuscript, we use the normal equation data from the ITSG-2016. The description of the data can be found in the data webpage provided in the manuscript (https://www.tugraz.at/institute/ifg/downloads/gravity-field-models/itsg-grace2016). As the derivation of the normal equation is not the focus of this study, we do not discuss it further but refer to the description in the data webpage and references therein (this is stated clearly in **Sect. 3.1** of the original manuscript). The element of the matrix A is mainly the partial derivative of the variational equation respect to the orbital information and gravitational coefficients. The variational equation include both orbital information and various kinds of L1B data including KBR data.

The accuracy of recovery has not been justified, which is essential for any scientific appliction to show that the accuracy of software is comparable with official products. Please include comparisons with the official ITSG2016 monthly solutions.

**R6:** The objective of this study is to combine GRACE normal equation with land surface model result to improve the estimated SM and GWS, and do not independently resolve the GRACE solution. We already presented the validation of SM and GWS computations. This is stated clearly in the introduction lines **80 – 82** of the submitted manuscript:

"The approach optimally combines the GRACE's least-squares normal equations with CABLE to improve ΔSM and ΔGWS estimates."

Therefore, the accuracy of the estimated result is only compared with the model-only result (please see **Fig. 6**). The accuracy of our TWS estimate can reach < 2 cm, which is in line with the GRACE accuracy of ~ 2 cm globally (Wahr et al., 2006).

Results of the inversion might be compared with those that assimilate GRACE into hydrological models to improve the surface/sub-surface storage compartments. Recent studies over Australia include: DOI:10.1002/2016WR019641; DOI:10.1016/j.advwatres.2017.07.001

**R7:** We thank for reviewer's suggestion. We already cited Tian et al. (2017) and approached the same conclusion about the benefit of GRACE on the groundwater estimate, and its limitation of the surface soil moisture.

**New references added**

[revised manuscript text omitted]

We so far discussed the GC approach to update the water storage estimates independently every month. The approach can be easily expanded to sequentially update the model initial states whenever the GRACE observation is available (for example, every day) as in data assimilation (DA) like ensemble Kalman filter (Evensen, 2003) and particle filter (Weerts and El Serafy, 2006). We briefly describe a way of modifying the GC approach suitable for DA. The ensemble of simulated monthly water storage estimates is predicted based on the set of ensemble forcing data and model parameters. This is simply running CABLE for $K$ (number of ensemble) times. When GRACE observation is available, the updated state is computed: ¶
$\hat{h}_{Re} = (\mathbf{C}_R + \mathbf{N}_R)^{-1}(\mathbf{C}_R \tilde{h}_{Re} + c_{Re}) \cdots (21)$¶
where the subscript $e$ represents the ensemble or perturbed version of the original vector or matrix (see e.g., Eq. (11)). The dimension of $\hat{h}_{Re}$, $\tilde{h}_{Re}$, $c_{Re}$ is $3J×K$. The estimated $\hat{h}_{Re}$ can be directly used as in the initial state for the next time step for CABLE run (Eicker et al., 2014; Tangdamrongsub et al., 2015; Tian et al., 2017), or used in the repeated run to avoid any spurious jump of the water storage estimates between the each step (Forman et al., 2012; Tangdamrongsub et al., 2017). This sequential update process can be carried out as long as desired. The feasibility of GRACE DA has been demonstrated with "devised" uncertainty (covariance) information. As a future work, we will develop new DA approach on the basis of full error information of GRACE data by using the least-squares normal equation and thus carrying the error information from the fundamental (satellite tracking) data level. ¶
¶

continual dry condition, leading to a greater decreasing of groundwater recharge and storage
over Western Australia.

The land surface model we used is deficient in anthropogenic groundwater consumption. The
model calibration will never help and the groundwater consumption must be brought in by
external sources. On the contrary, the statistical approach like our GC approach may be
useful to fill in the missing component and lead to a more comprehensive water storage
inventory.

However, it is difficult to constrain different water storage components by only using total
storage observation like GRACE. In addition, it is challenging to improve surface soil
moisture varying rapidly in time, using a monthly mean GRACE observation. Tian et al.
(2017) utilized the satellite soil moisture observation from the Soil Moisture and Ocean
Salinity (SMOS, Kerr et al., 2001) in addition to GRACE data for their data assimilation and
showed a clear improvement in the top soil moisture estimate. The GC approach with
complementary observations at higher temporal resolution should be considered particularly
to enhance the surface soil moisture computation.

Finally, the GC approach can be simply extended for GRACE data assimilation. Assimilating
the raw GRACE data into land surface models like CABLE enables the model state and
parameter to be adjusted with the realistic error information, allowing reliable storage
computation. The GC data assimilation will be developed in our future study.

**Acknowledgement**

This work was funded by NASA projects on GRACE and GRACE Follow-On missions and
partly by Australian Research Council (DP160104095). MD was supported by ARC Centre
of Excellence for Climate Systems Science. HK was supported by Japan Society for the
Promotion of Science KAKENHI (16H06291). We thank Torsten Mayer-Gürr for GRACE
data products in the form of the least-squares normal equations. We also thank three
anonymous reviewers for helping us improve the manuscript.

[revised manuscript text omitted]

---

## Author Response (AR2)

**Referee #1**

I read once more a version of the paper and also the rebuttal letter in response to my first review. However, I am not satisfied with the provided answers on two major points raised in my first review. I am therefore recommending to ask the authors for another revision in order to give them the chance to properly address those concerns.

We thank reviewer 1 for the comments. Below are our responses (in blue) to new comments #1 and #2, including the original comments and replies.

Major Comment # 1

(Original review) I suggest that comparisons with the official ITSG2016 monthly solutions are included in order to demonstrate the added-value of the GC approach over the standard L2 data. Note that comparisons against GRGS or JPL monthly solutions as already (partly) included in the paper will not be sufficient since ITSG2016 is commonly perceived as a GRACE series of particularly high quality.

(Reply) The computation of Delta TWS from ITSG L2 solutions (like other L2 solutions) is subject to the post-processing filters and often followed by the application of empirical scaling factors, obtained from the land surface models. Our study provides a more rigorous way of computing Delta TWS without going through such ad hoc procedures. GRGS and JPL mascon are internally or post-processed and it is not expected users to apply (subjective) post- processing. This is the primary reason we validate our results with GRGS and JPL mascon.

(Second review) I disagree. The results of the new method needs to be compared exactly to conventionally post-processed gravity fields of the very same L2 data source in order to demonstrate the superiority of the proposed method. Conventional postprocessing at least implies restoring spectral deficits (degree 1, 2) and the removal of correlated errors (Kusche et al., 2009; or Swenson and Wahr, 2006). From my point of view, those are in no way just "ad hoc procedures".

(Reply) We added the comparison with the ITSG solutions in the Figure 6 of the revised manuscript. The revised Figure 6 now compares the ΔTWS results from ITSG (a), JPL/Mascon (b), and GRGS solutions (c), and this study GC (e). This particular example of ITSG is from the DDK5 filtering in addition to usual replacement of degree-1 and degree-2/order-0 coefficients. For the reviewer's information, we demonstrate how the different filters (for example, DDK1, DDK5, and DDK8 of Kusche et al., 2009) affect the computation of the storage in Figures S1 and S2.

[Figure]

**Figure S1:** ΔTWS of April 2003, derived from the same ITSG solution with different post-processing filters (DDK1, DDK5, and DDK8) applied.

[Figure]

**Figure S2:** Basin average ΔTWS of the North East Coast basin computed from ITSG-DDK1 and ITSG-DDK8 solutions. The RMS difference between two results is ~ 5.3 cm, greater than the amplitude of DDK1.

Major Comment # 2

(Original review) The GC approach assumes that model errors are normally distributed with zero mean (eq. 1). Authors should provide more evidence that this assumption is indeed justified in their setting.

(Reply) R3: As the reviewer concerned, the GC approach is developed based on the least-square combination, which assumes the uncertainty following the normal distribution with zero mean and covariance C. The derivation and setting of model uncertainty under the given assumption (e.g., zero mean) and its limitation are clarified in Sect. 4.2.
"The GC approach assumes that model errors are normally distributed with zero mean. Any violation of this assumption will yield a bias in the combined solutions. Therefore, the mean value is removed from each ensemble member, $\mathcal{H}_R' = \mathcal{H}_R - \widetilde{h}_R$, and the error covariance matrix of the model is empirically computed as"

(Second review) My original request was to provide evidence that the assumption of normal distribution is indeed justified. This was by no means attempted by the presented changes to the manuscript.

(Reply) We thank the reviewer 1 for being patient to clarify the comment once again. The distribution of the model errors is demonstrated in Figure S3 in this letter as well as Figure 2 in the revised manuscript. The figure illustrates the histogram of the 210 ensemble members ($\mathcal{H}_R$) for $\Delta \mathrm{SM}_{\mathrm{top}}$, $\Delta \mathrm{SM}_{\mathrm{rz}}$, and $\Delta \mathrm{GWS}$, after removing the respective means ($\widetilde{h}_R$). The normal distribution of the model error used in the GC approach is grounded on the distribution of $\mathcal{H}_R'$ ($\mathcal{H}_R' = \mathcal{H}_R - \widetilde{h}_R$) or the histogram of the ensembles. We revised the manuscript and added this new Figure in Section 4.2.

[Figure]

**Figure S3:** Histograms of the model errors ($\mathcal{H}_R{}'$) computed from 210 ensemble members without the mean. The basin averaged values (from all 10 Australian basins) of January 2003, for example, are used to compute the histogram.

**Referee #3**

This study introduces an inversion technique to use GRACE L1B data to improve the estimation of soil moisture and groundwater within Australia. As I mentioned in the previous round, I believe this line of research is interesting, however, I am not convinced that the proposed technique is well descried (with the current formulation it is not possible to re-do the work), and also it is not well justified (validation does not prove that the new method works any better than other available techniques). I recommend a reject / major review decision for this contribution.
We thank the reviewer 3 for the comments. Below is our responses (R, in blue):

Major Concerns:
A number of my previous comments are not adequately addressed
• The Methodology section needs to be specified, please add appendices to clearly how the equations are built. I cannot figure out how the normal equation is formulated, whether it includes KBRR and any orbital information? L120–> Please describe how the matrix A is derived and what are the entries. Similarly L128-L130 are unclear.
R1: As written in the original and revised manuscripts, we use the monthly least-square normal equation data from ITSG that are built upon GRACE L1B data products including inter-satellite ranging data. To clarify this further, we revised Section 2 of the manuscript by highlighting the use of the normal equations and moving the observation equation part to the appendix. The GRACE normal equation data are available from, as already given in the manuscript:
https://www.tugraz.at/institute/ifg/downloads/gravity-field-models/itsg-grace2016

• The accuracy of recovery has not been justified a synthetic study should be added to show the framework recovers the introduced gravity signals with a sufficient accuracy, see e.g., https://academic.oup.com/gji/article/158/3/813/2062077
• I cannot accept the following argument: "The approach optimally combines the GRACE's least-squares normal equations with CABLE to improve ΔSM and ΔGWS estimates." Therefore, the accuracy of the estimated result is only compared with the model-only result (please see Fig. 6). The accuracy of our TWS estimate can reach < 2 cm, which is in line with the GRACE accuracy of ~ 2 cm globally (Wahr et al., 2006).
R2: We are afraid that if we understand what is asked by this comment. Of course, our method and numerical codes were already tested with synthetic data sets, before applying it to the real data. We have verified the accurate recovery (within the numerical precision) from the synthetic data. This should be clear from our least-squares development as detailed in the manuscript. In addition, the comprehensive accuracy estimates of different water storage components is presented in Figure 7 of the revised manuscript.

A large part of the introduction is used to state other methods (e.g., assimilation and decomposition-based inversion) are erroneous. If this is true, reliable evidences must be provided to indicate the presented method works better than other technique
R3: The statements and related references regarding to signal decomposition were removed. For data assimilation, the evidences can be found in the references given (e.g., Girotto et al., 2016; Tian et al., 2017). In this introduction section, we only provide the general background of how ΔGWS has been estimated from GRACE, we do not claim that our presented method works better than other technique.

• L16: (e.g., time-variable gravity fields, i.e., Level 2 data, and ...
R4: The statement is modified.

• L21 remove ', not the post-processed ΔTWS grid data.'
R5: Removed.

• L27: ....combination maximizing the strength of the model and observations while suppressing the weaknesses. The approach .....   This is not well justified, in other words, the authors do not show combining the final grace products and models is worse than the proposed joint-inversion.

R6: Revised. "The GRACE-combine (GC) approach is developed for optimal least-squares combination and the approach is applied to estimate…".

• L33-34: ... estimates likely due to limitation of GRACE's temporal and spatial resolution... The way this has been validated in this study does not necessarily back this conclusion up.
R7: Revised. "Significant improvement is found in groundwater storage while marginal improvement is observed in surface soil moisture estimates".

• GRACE information rigorously and negate these limitations, this study uses the fundamental
R8: The statement is incomplete, and we do not take any action on this comment.

• L48-L51, Also L55, 59,etc.: before the references add 'e.g.,' as these references are only examples of existing researches.
R9: Done.

• L61: (Schumacher et al., 2016,2018; Tangdamrongsub et al., 2017 )
Schumacher, M, Kusche, J & Döll, P, 2016, A systematic impact assessment of GRACE error correlations on data assimilation in hydrological models. Journal of Geodesy, vol 90., pp. 537-559
Schumacher, M, Forootan, E, van Dijk, A, Schmied, HM, Crosbie, R, Kusche, J & Döll, P, 2018, Improving drought simulations within the Murray-Darling Basin by combined calibration/assimilation of GRACE data into the WaterGAP Global Hydrology Model. Remote Sensing of Environment, 204, pp212-228, doi:10.1016/j.rse.2017.10.029
R10: References are added.

• L80 'incorrect gravity information' --> does it mean all previous researches are wrong?
R11: The application of data tuning alters the original gravity information. This is self-explained.

• L81 ' began to employ' --> have applied the L2's (Schumacher etl al., 2016, 2018; Khaki et al., 2017 a,b)
Khaki, M., Hoteit, I., Kuhn, M., Awange, J., Forootan, E., van Dijk, A., Schumacher, M., Pattiaratchi, C. (2017a), Assessing sequential data assimilation techniques for integrating GRACE data into a hydrological model. Advances in Water Resources, 107, pages 301-316, doi:10.1016/j.advwatres.2017.07.001
Khaki, M., Schumacher, M., Forootan, E., Kuhn, M., Awange, J., van Dijk, A. (2017b), Accounting for spatial correlation errors in the assimilation of GRACE into hydrological models through localization. Advances in Water Resources, 108, pages 99-112, doi:10.1016/j.advwatres.2017.07.024
R12: References are added.

• L83 'still affected by the post-processing filter' --> This statement is very negative, and criticizes other processing strategies without providing a real measure. As far as I understand, in an assimilation formulation, incorporating the full co-variance matrix is already very important, useful and sufficient. Other post-processing steps have less impacts on the final assimilation results. See Schumacher et al 2018 and discussions for the Australian case study. If the authors suggest their approach is better than other formulations of signal separation, it should be validated and discussed.
R13: Revised. "Some recent studies began to employ the full variance-covariance information in the data assimilation scheme to enhance the quality of the estimates (Eicker et al., 2014, Schumacher et al., 2016; Tangdamrongsub et al., 2017; Khaki et al., 2017 a,b)".

• L86: It is not clear why this CABLE model has been selected
R14: CABLE provides comprehensive water storage component suitable for our analysis, and particularly the code and model parameter are publicly available. The additional sentence is added to the text to clarify this: "CABLE is a public available land surface model, and can be used to estimate soil moisture and groundwater in terms of volumetric water content …".

• After L92: (Dis-)Similarity of this work with previous studies that use inversion techniques might be addressed, see e.g., https://academic.oup.com/gji/article/158/3/813/2062077
R15: We thank for reviewer suggestion. The reference is addressed in the revised manuscript.

• Please also report on the impact of choices, within the gravity inversion, on the final mass estimation see e.g., http://www.sciencedirect.com/science/article/pii/S0264370716301016?via%3Dihub
R16: It is not clear what the reviewer means by this comment. There are various GRACE processing results around including the one the reviewer refers. It is not the focus of comparing all kinds of GRACE processing in this study, but demonstrating the way of using GRACE data with the actual covariance (normal equation) information.

• After L98: You might add a discussion reflecting the fact GRACE data is sensitive to many processes. You select the signal over Australia to simplify the computation etc.
R17: Revised. "One advantage of the study area is that the state vector can be defined mainly by soil moisture and groundwater as other hydrological components (e.g., glacier) are negligible.".

• L109: 'from a model' -->what does this mean? which model?
R18: Revised. "a land surface model".

• Equation5 --> matrices should be shown, e.g., add a appendix
R19: Revised. Please see R1.

• L162: How is the contribution of the signal from outside removed or corrected?
R20: The contribution of the signal from outside is removed using the GRGS solution. This was clearly described in Sect. 2 and Sect. 3.2 of the revised manuscript.

• Equation11 --> Please discuss the condition of these matrices used in the inversion.
R21: The matrix is well conditioned, and the condition number is around $10^4 – 10^5$.

• Co-variance estimation of the solution and whether it is representative of all measurements errors and model errors should be added.
R22: Fundamentally, the errors estimated from the least-squares combination present both model and observation errors (please see Eq. (7)). We present the covariance estimates (in terms of standard error) in Fig. 7 of the revised manuscript.

• 4.2 Model uncertainty --> From a hydrological point of view, model's errors should contain those uncertainties related to parameters, forcing data and model structure. The current errors do not reflect all these three categories and even the assumptions, that are used to define the distribution of model parameters, are not that sophisticated. Therefore, the impact of over-/under-estimation of the model's co-variance matrix on the final inversion results must be evaluated.
R23: In this study, the model error contains uncertainties related to parameters, forcing data and model structure, as the reviewer mentioned. This is written very clearly in Sect. 4.2. In this paper, the model forcing data (mainly precipitation) and parameters are both perturbed. We estimate the precipitation error based on 7 different products, which we believe it provides more realistic error compared to a simple assumption (e.g., 10-30 % of the value) seen in the previous publications and current practices (e.g., Eicker et al., 2014; Tangdamrongsub et al., 2015). The offline sensitivity study of forcing data is also conducted, and it is found that the water storage estimate is most sensitive to precipitation data, and relatively less sensitive to the change of other forcing components (this is written in Sect. 4.1). This is the main reason the precipitation is mainly perturbed. The parameters are perturbed based on the recommendation of the previous literature and the omission error is also included. In fact, most of previous literature (e.g., Zaitchik et al., 2008; Forman et al., 2012; Eicker et al., 2014; Tian et al., 2017, etc.) adopt a very similar procedure of model error determination we use here.

[revised manuscript text omitted]

or

$$\mathbf{N}_R\hat{\boldsymbol{h}}_R = \boldsymbol{c}_R \qquad\qquad (8)$$

where $\mathbf{N}_R = \mathbf{H}_R^T\mathbf{Y}_R^T\mathbf{N}\mathbf{Y}_R\mathbf{H}_R$ and $\boldsymbol{c}_R = \mathbf{H}_R^T\mathbf{Y}_R^T\boldsymbol{c} - \mathbf{H}_R^T\mathbf{Y}_R^T\mathbf{N}\mathbf{Y}_o\mathbf{H}_o\hat{\boldsymbol{h}}_o$. As seen, Eq. (7) is the regional representation of Eq. (5) where only the grid cells inside the study region are used, while the contribution from the grid cells outside the region needs to be removed or corrected. Combining the normal equation of Eq. (2) and Eq. (8), the optimal combined solution of $\hat{\boldsymbol{h}}_R$ can be resolved as follows:

$$\hat{\boldsymbol{h}}_R = \left(\mathbf{C}_R^{-1} + \mathbf{N}_R\right)^{-1}\left(\mathbf{C}_R^{-1}\tilde{\boldsymbol{h}}_R + \boldsymbol{c}_R\right) \qquad (9)$$

The computation of model covariance matrix $\mathbf{C}_R$ will be discussed in Sect. 4.2. The posteriori covariance of $\hat{\boldsymbol{h}}_R$ can be estimated as follows:

$$\hat{\mathbf{\Sigma}} = (\mathbf{C}_R^{-1} + \mathbf{N}_R)^{-1}, \qquad\qquad (10)$$

and the uncertainty estimate of $\hat{\boldsymbol{h}}_R$ is simply calculated as:

[revised manuscript text omitted]

---

## Author Response (AR3)

**Response to editor**

We deeply appreciate the editor's patient of handling our manuscript, and below is our responses (**R**, blue) based on the comments

Therefore, while I find that this is an interesting new method that is worth publishing, it needs to be much clearer the manuscript is mainly presenting a new approach and not attempting to demonstrate its superiority.

**R1:** We agree with the editor's suggestion. In the revised version, we modify and remove all misleading statements regarding the superiority of the approach.

- The abstract should mention that the improvement provided by the GC approach is only over the CABLE model (last two sentences of the abstract).

**R2:** We revise the sentence as follow:

"Comparing to CABLE, we demonstrate the GC approach delivers evident improvement of water storage estimates, consistently from all basins, yielding better agreement at seasonal and inter-annual time scales."

We also revise the objective of the study to clarify this, as follows:

"The main objective of this study is to present the GRACE-combined (GC) approach to improve the model estimated $\Delta$SM and $\Delta$GWS at regional scales."

- Please ensure throughout the manuscript that it is clear that you are not claiming superiority of this method over other techniques (e.g. the term 'improved' should be removed from the title, because it suggests improvement over other methods).

**R3:** All misleading statements regarding the superiority of the approach are removed or modified. The term "improved" is also removed from the title.

- In the caption for figure 6, panel labels a-c should be introduced before labels d-k. I would therefore suggest changing the order of the panels in the figure (e.g. putting the top three panels at the bottom, and separating them from the other 8 panels; otherwise the reference to columns two and three is confusing).

**R4:** We thank for the editor's suggestion. We modify the figure based on the suggestion. Now, Fig. 6 only contains 8 panels, and the GRACE results are presented in a separated figure (Fig. 7).